# Peptidergic Systems and Cancer: Focus on Tachykinin and Calcitonin/Calcitonin Gene-Related Peptide Families

**DOI:** 10.3390/cancers15061694

**Published:** 2023-03-09

**Authors:** Manuel Lisardo Sánchez, Francisco D. Rodríguez, Rafael Coveñas

**Affiliations:** 1Laboratory of Neuroanatomy of the Peptidergic Systems, Institute of Neurosciences of Castilla and León (INCYL), University of Salamanca, c/Pintor Fernando Gallego 1, 37007 Salamanca, Spain; lisardosanchez8@gmail.com; 2Department of Biochemistry and Molecular Biology, Faculty of Chemical Sciences, University of Salamanca, 37008 Salamanca, Spain; lario@usal.es; 3Group GIR-USAL: BMD (Bases Moleculares del Desarrollo), University of Salamanca, 37008 Salamanca, Spain

**Keywords:** neurokinin, NKA, NKB, CGRP, amylin, adrenomedullin, adrenomedullin 2, peptide, peptide receptor

## Abstract

**Simple Summary:**

Neurokinins A and B, adrenomedullin, adrenomedullin 2, amylin, and calcitonin gene-related peptide are essential in different tumors. These peptides are involved in tumor cell proliferation and migration, metastasis, angiogenesis, and lymphangiogenesis. Accordingly, several antitumor therapeutic strategies, including peptide receptor antagonists, can be developed. This review highlights the essential roles played by both tachykinin and calcitonin/calcitonin gene-related peptide families in cancer progression, which support the application of promising clinical antitumor therapeutic strategies.

**Abstract:**

The roles played by the peptides belonging to the tachykinin (neurokinin A and B) and calcitonin/calcitonin gene-related peptide (adrenomedullin, adrenomedullin 2, amylin, and calcitonin gene-related peptide (CGRP)) peptide families in cancer development are reviewed. The structure and dynamics of the neurokinin (NK)-2, NK-3, and CGRP receptors are studied together with the intracellular signaling pathways in which they are involved. These peptides play an important role in many cancers, such as breast cancer, colorectal cancer, glioma, lung cancer, neuroblastoma, oral squamous cell carcinoma, phaeochromocytoma, leukemia, bladder cancer, endometrial cancer, Ewing sarcoma, gastric cancer, liver cancer, melanoma, osteosarcoma, ovarian cancer, pancreatic cancer, prostate cancer, renal carcinoma, and thyroid cancer. These peptides are involved in tumor cell proliferation, migration, metastasis, angiogenesis, and lymphangiogenesis. Several antitumor therapeutic strategies, including peptide receptor antagonists, are discussed. The main research lines to be developed in the future are mentioned.

## 1. Introduction

To attain a better quality of life, with higher cure rates and fewer sequelae in cancer patients, new molecular targets and compounds that specifically destroy tumor cells must be urgently investigated. One of these promising targets could be peptidergic systems, i.e., peptides and their receptors, which play a crucial role in cell communication. Peptidergic systems have opened up new research lines and possibilities to improve cancer diagnosis and explore new antitumor strategies [1,2,3,4]. The involvement of peptidergic systems in cancer has attracted increasing interest in the last few years. Peptides such as substance P (a full review focused on its participation in cancer was recently published [5]), neurotensin, orexin, angiotensin II, neuropeptide Y, vasoactive intestinal peptide, calcitonin gene-related peptide, adrenomedullin, adrenomedullin 2 or intermedin, and amylin contribute to cancer development [1,2,6]. These peptides promote the mitogenesis/migration of tumor cells, exert an antiapoptotic action, and stimulate the growth of blood vessels and lymphangiogenesis. However, some peptides such as neuropeptide Y, orexin, and vasoactive intestinal peptide also exert an anticancer effect [7]. Furthermore, tumor cells release peptides acting through autocrine, paracrine, and endocrine (tumor mass) mechanisms [1,2,3,5,6,8,9,10,11]. While some peptides (e.g., the heptapeptide angiotensin (1-7)) exert an anticancer effect, others (e.g., galanin) promote both proliferative and antiproliferative actions on tumor cells [3].

Tumor cells also overexpress peptide receptors, allowing for a specific treatment against cancer cells with peptide antagonists. In addition, the overexpression of these receptors can be used as a prognostic biomarker [1]. The overexpression of peptidergic systems has been associated with higher tumor aggressiveness, tumor size, poor prognosis, worse sensitivity to chemotherapy agents, and increased relapse risk [1,5]. Peptide antagonists promote apoptosis in tumor cells, block the migration of cancer cells, and inhibit angiogenesis. In combination therapy with chemotherapy, peptide antagonists decrease the side-effects promoted by cytostatics and exert a synergic effect [1,4]. Because peptide receptors are potential new targets in cancer treatment, peptide antagonists are promising antitumor drugs. Because the involvement of substance P in cancer through the neurokinin-1 receptor is widely known [5], this review aims to update the findings regarding the involvement of the main peptides belonging to the tachykinin (neurokinin A and B) and calcitonin/calcitonin gene-related peptide (adrenomedullin, adrenomedullin 2, amylin, and calcitonin gene-related peptide) systems in the mitogenesis, migration, and invasion of tumor cells, angiogenesis, and lymphangiogenesis; moreover, according to the data reported, this review suggests anticancer therapeutic strategies targeting tachykinin and calcitonin peptidergic systems. The signal transduction pathways, mediated by the different tachykinin/calcitonin receptors involved, are reviewed, along with the structure and dynamics of these receptors.

Currently, it is considered that the tumor microenvironment is composed of tumor cells and cancer stem cells, as well as normal stromal cells. Hence, tumors are not currently regarded as a simple mass of cancer cells [12,13]. Updated hallmarks of cancer cells escaping from normal behavior are the following: (1) growth suppressor evasion; (2) proliferative signaling maintenance; (3) replicative immortality; (4) cell death resistance; (5) invasion/metastasis activation; (6) angiogenesis promotion; (7) inflammation; (8) genome instability; (9) energy metabolism reprogramming; (10) immune destruction evasion; (11) senescent cells; (12) polymorphic microbiomes; (13) non-mutational epigenetic reprogramming; (14) unlocking phenotype plasticity [12,13] (Figure 1). Tachykinin peptides are involved in five of these hallmarks, whereas those belonging to the calcitonin/calcitonin gene-related peptide (CGRP) family are involved in nine. This comment shows the essential roles of both peptidergic systems in cancer development and the numerous research lines that can be developed.

## 2. Tachykinin and Calcitonin Peptide Families

### 2.1. Tachykinin Peptide Family

The tachykinin family of peptides includes kassinin, ranakinin, eledoisin, neuropeptide K, hemokinin-1, substance P (SP), neurokinin B (NKB), and neurokinin A (NKA) [14]. Activated metabotropic neurokinin receptors (neurokinin-1 receptor (NK-1R), NK-2R, and NK-3R), widely distributed by the central and peripheral nervous systems, exert many physiological actions and determine many pathophysiological [15,16,17]. NKA and NKB mediate many physiological effects and are involved in several pathologies [16,18,19,20,21,22].

#### 2.1.1. Genes and Products of Human Tachykinins

Tachykinin is the general name for a large family of peptides (hundreds) found in all bilaterians (from insects to amphibians and humans) [23,24]. In humans, three principal tachykinins, SP, NKA, and NKB, participate in cellular mechanisms responsible for physiological and pathological outcomes. Three human genes named *TAC1*, *TAC3*, and *TAC4* encode the peptides mentioned above, plus hemokinin1 (HK-1) and endokinins [17,25,26,27]. Tachykinin genes transcribe into splice variants [28,29,30]. Gene *TAC1* with seven exons is located on chromosome 7 (7q21.3). It gives rise to four different splice mRNA variants, α, β, γ, and δ, that encode four protachykinin peptide isoforms of various sizes, α (111 residues), β (129 residues), γ (114 residues), and δ (96 residues). Exon 3 of TAC1 appears transcribed in all four mRNA variants. It encodes the sequence of the undecapeptide SP. The exon 6 sequence transcript is only present in the β and γ splice variants and encodes the decapeptide NKA. Gene *TAC3*, with seven exons, is on chromosome 12 (12q13.3) and generates two alternative coding transcript variants, isoforms 1 and 2. Isoform 1 translates into a peptide precursor of 121 residues, and isoform 2 translates into a peptide precursor of 103 amino acids. Exon 5 of both mRNAs encodes the decapeptide NKB. Gene *TAC4* sits on chromosome17 (17q21.33) and encodes five splice variants, α, α-2, β, γ, and δ, translating into five peptide precursors that provide functional HK-1 and endokinins after post-translational processing (Figure 2).

Tachykinin peptide precursors undergo post-translational modifications, generating functional peptides [31]. Endopeptidase processing at a specific pair of basic residues liberates the N-terminal ends of SP, NKA, and NKB from their respective precursors. SP occupies amino-acid positions 58–68 in all precursor peptide variants encoded by TAC1. NKA corresponds to amino-acid positions 98–107 in the precursor variants β and γ. Lastly, precursor peptides 1 and 2 encoded by the *TAC3* gene contain NKB (amino-acid positions 81–90). Splitting the C-terminal ends of the three neurokinins from their protachykinins follows a shared mechanism required for functional activity, consisting of a PAM (peptidyl glycine α-amidating monooxygenase, EC 1.14.17.3)-catalyzed reaction cleaving the N–Cα bond between methionine and neighboring glycine [32,33] (Figure 2). The reaction’s products are the C-terminal amidated active neurokinin and an N-glyoxylated peptide.

**Figure 2 cancers-15-01694-f002:**
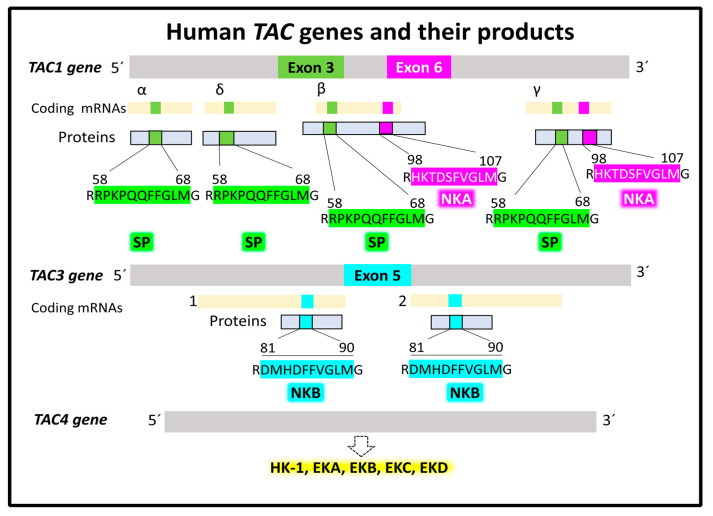
Human tachykinin genes and their main products, SP, NKA, and NKB. Alternative splicing generates several mRNA isoforms transcribed into protein products that undergo post-translational modifications (protein hydrolysis and amination of C-terminal methionine), giving rise to functional tachykinins. Data supporting this scheme are from the UNIPROT [25] and Entrez Gene (National Library of Medicine) [34] databases.

#### 2.1.2. Structure of NK-2R and NK-3R

Three mammalian neurokinin receptors convey the activity of tachykinins. NK-1R binds with high-affinity SP and HK-1, NK-2R preferentially binds NKA, and NKB is the natural agonist of NK-3R. All three neurokinins exhibit full agonist capacity in all three receptors, albeit with different rank order potency [35]. Figure 3 shows the primary and general tertiary serpentine structure of NK-2R and NK-3R, together with the amino-acid sequence of their preferred endogenous agonist ligands. Sequence homology between NK-2R and NK-3R is 57%. NK-2R has a sequence homology of 54% with NK-1R; NK-3R shares a sequence homology of 61% with NK-1R (according to the SIM alignment tool [36]). The tertiary structure of the NK-2R and NK-3R is very similar when predicted with Alpha-Fold (Figure 4 and Figure 5). However, specific residue positions and transmembrane segment displacements make interactions with agonists and antagonists different. Neurokinin receptors belong to class A of the large family of G-protein-coupled receptors (GPCR), membrane-bound proteins sharing a 3D structure arranged in seven transmembrane domains linked with extra and intracellular loops [37,38]. Structural and functional studies of these membrane proteins have provided broad information on the structure–activity relationships that explain their role in cell function and their relevance in drug discovery with therapeutic applications [38,39,40,41,42]. Some recent research and reviews provided detailed and sound information on the structure and dynamics of NK-1R [43,44,45,46]. This section focuses on the architecture of the human NK-2R and NK 3R.

##### The Structure of NK-2R

NK-2R is a monomeric integral membrane protein (UNIPROT, P-21452) [25] made of 398 amino-acids that expand through the plasma membrane lipid bilayers with seven domains (Figure 3). The human gene *TACR1* (Gene ID 6865 from the Entrez Gene database, National Library of Medicine [34]) has five exons with chromosomal localization 10q22.1 and encodes the protein NK-2R [49]. Post-translational modifications of NK-2R include glycosylation of asparagines 11 and 19 (N-terminal region), a disulfide bridge between cysteines 106 (in TM3) and 181 (in ECL2), and the palmitoylation of Cys324 (intracellular C-terminal domain). The preferred endogenous ligand of NK-2R is NKA (Figure 3 and Figure 4).

Early pharmacological studies in isolated organs revealed the functional importance of the NKA sequence from amino acids 4–10. This sequence retained the activity of full NKA (1-10) [50]. Additionally, the substitution of Gly8 for Ala8 increased the selectivity and potency of the short form of NKA [51] Radioligand binding and functional experiments in human specimens showed the importance of amino acids Asp4, Phe6, Val7, Leu9, and Met10 for binding selectivity [52]. The position of Phe6 within the C-terminal pentapeptide determines the binding of NKA to all neurokinin receptors [53]. It makes important contact with the protein through aromatic (π–π) and amino–aromatic (N–π) interactions [50]. Site-directed mutagenesis bestowed crucial amino-acid positions in transmembrane helices 3, 5, and 7 as part of the binding site for NKA and different interactions with SP and NKB agonists, and with NK-2R antagonist SR-48968 [54]. Homology modeling analysis implemented indirect data to attain forms of minimal energy to understand the coupling of agonists, antagonists, and cellular signaling proteins. Molecular modeling and docking of the NKA within NK-2R, using rhodopsin as a structural template and the three-dimensional NKA structure determined with NMR [55] (Figure 4) resulted in the definition of the site where NKA contacts the receptor protein. Within a distance of less than 3 Å, predicted hydrogen bonds (residues His1, Lys2, Thr3, Asp4, Gly8, Leu9, and Met10) stabilize the agonist and receptor interaction. Furthermore, residues Phe6, Leu9, and Met10 are relevant in biological activity. Furthermore, in the established model, some interactions anchor the binding pocket for the agonist. Figure 4 depicts the atomic contacts between the C-terminal portion of NKA and NK-2R. For example, the amidated methionine at the NKA’s C-terminal end contacts several residues of the NK-2R (Ala116, Trp263, Met117, and Ser298). The phenyl ring of Phe6 sits in a hydrophobic cavity employing π–π interactions with residues bearing aromatic rings (Tyr266 and Phe270) and other groups; Leu9 connects through weak interactions with the lateral chains of Ile114, Met117, Ala116, Trp263, and Tyr266. The conserved C-terminal pentapeptide sits in a hydrophobic cavity built by transmembrane helices TM2, TM3, TM6, and TM7 [56] (Figure 4C). The building of a model of NK-2R bound to NKA by comparing sequences with the rhodopsin receptor and analysis of fluorescence resonance energy transfer (FRET) with fluorescent NKA revealed that the N-terminal end of NKA expands to the extracellular medium. In contrast, the C-terminal amidated end is buried in the TM-spanning domains [57]. Site-directed mutagenesis analysis of the antagonist binding site of NK-2R showed that nepadutant (a peptide) partially overlapped with the aperture occupied by the non-peptide antagonist SR-48968. Aromatic residues in TM5, TM6, and TM7 play a fundamental anchor function to accommodate the antagonists studied [58].

NK-2R may adopt different active conformations where NKA adapts to generate different intracellular responses (biased agonism). Different receptor conformations linked to cellular responses facilitate the design of allosteric modulators that may help to control specific signaling by affecting the affinity of NKA to the receptor’s conformations [59]. Binding selectivity depends on the C-terminal end and the interactions of amino acids of the rest of the molecule with the receptor. A recent cryo-EM structure determination of NK-2R-Gq protein complex bound to NKA [60] underlies the importance of several residues that secure the position of NKA within de receptor pocket (see Figure 4D). Lateral chains of residues Y93 and N90 and N97 in TM2 interact with the backbone carbonyl oxygens of Val7 and Leu9. The backbone nitrogen atom of Gly8 forms a hydrogen bond with the hydroxyl group of Y289 in TM7.

Further stabilization of M10 occurs with I114 in TM3 through hydrophobic interaction. Additionally, the N-terminus of NKA contacts the ECL2 region of NK-2R (D175 forms a salt bridge with Lys2 carbonyl oxygen), and Phe6 further contacts Met28 in the N-terminal part of NK-2R and I285 in TM7 (Figure 4D). When Sun et al. [60] compared NK-2R with NK-1R (PDB ID 7RMG, [61]), they found very similar structures, except for an outward shift of the TM5 in NK-2R of 2.4 Å. Moreover, they identified significant differences in the arrangement of extracellular loops, specially ECL2. The differences may explain endogenous ligand selectivities for the three receptors, notably concerning the interaction of the N-terminal sequence of neurokinins with the extracellular loop 2. Nevertheless, we need additional architectural and structure dynamics studies (X-ray crystallography, cryo-electron microscopy, and NMR) to ascertain the atomic environment that accommodates agonists, antagonists, and allosteric modulators. The nuanced structural analysis will unravel the role of these receptors in physiological and pathological outcomes and their collaboration with other NK receptors responding to the three agonists, NKA, SP, and NKB, with different selectivity.

**Figure 4 cancers-15-01694-f004:**
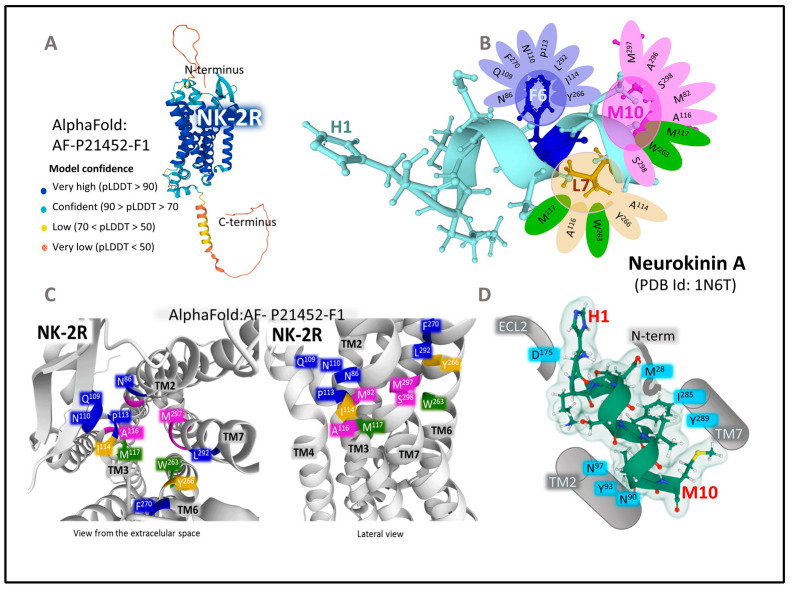
(**A**) represents the predicted structure of NK-2R (UNIPROT P21452) according to the Alpha-Fold prediction [62,63,64]. The parameter pLDDT (predicted local distance difference test) assessed the model confidence. (**B**) depicts the 3D structure of NKA determined by NMR obtained from the Protein Data Bank [65] (PDB ID 1N6T [55]) drawn with Mol* free web-based software [48]. According to the model, the contacts of NKA’s C-terminal amino acids with the receptors’ amino acids are depicted. (**C**) shows two views of a three-dimensional representation of NK-2R, obtained from modeling and docking studies [56], indicating the situation of principal amino-acid positions in the Alpha-Fold model of NK-2R closely contacting the C-terminal region of NKA. Residues in green contact more than one pharmacophore of NKA. (**D**) represents further contacts of NKA (PDB ID 1N6T, [56]) with NK-2R according to data from the cryo-EM structure provided by Sun et al. [60].

##### The Structure of NK-3R

NK-3R is a monomeric membrane protein made (UNIPROT P29371) [25] of 465 amino acids that expand through the plasma membrane lipid bilayers with seven domains (Figure 3). The human gene *TACR3* (Gene ID 6870), [34] has five exons with chromosomal localization 4q24 and encodes the protein NK-3R [66]. Main post-translational modifications of the receptor include the glycosylation of asparagines 23, 50, and 73 (in the N-terminal region), a disulfide bridge between cysteines 158 (in TM3) and 233 (in ECL2), and the palmitoylation of cysteine at position 374 in the intracellular C-terminal domain. The preferred endogenous ligand of NK-3R is NKB (Figure 3 and Figure 5).

Molecular modeling studies using the structure of rhodopsin receptors and manual docking of NKB (PDB ID 1P9F [67] within the NK-3R model (Figure 5) resulted in the definition of the site where the NKB C-terminal region contacts the receptor protein [68]. According to the refined model, the C-terminal region of NKB lodges three pharmacophores (Phe6, Leu9, and Met10) that stabilize closely interacting with three hydrophobic holes in the NK-3R. The aromatic ring of Phe6 favors contacts with residues S130, P165, I166, A168, V169, F170, and W312. Interactions π–π between the phenyl group of F6 and the aromatic rings of F170 and W312 orientate the peptide in an optimal position within the receptor cleft. The isobutyl l chain of L9 contacts with residues C311, W312, P314, L344, A345, M346, and S347. The binding pocket for Met10 includes residues F123, V169, S172, M176, V304, F308, C311, S347, S348, and M350. (Figure 5B,C).

Docking analysis performed in a model of NK-3R based on the 3D structure of bovine rhodopsin reported different binding sites for the NKB in TM domains 2, 6, and 7, as well as ECL2. Additionally, the antagonists metaltenant and osatenant occupied different positions within the receptor, indicating that the binding site for the antagonists greatly coincided but did not entirely overlap [69]. Using a model based on the structure of bovine rhodopsin, Geldenhuys et al. [70] proposed a pharmacophore for quinoline derivatives with antagonistic properties formed by three groups: two aromatic rings, two hydrogen donors, and one aromatic acceptor that would anchor in a cleft within the receptor structure.

Further refined structural analysis will unravel the atomic setting accommodating agonists, antagonists, and receptor modulators. The structure dynamics analysis will contribute essential information supporting the physiological and pathological contribution of NK-3R and their collaboration with other NK receptors responding to the three agonists, NKA, SP, and NKB.

**Figure 5 cancers-15-01694-f005:**
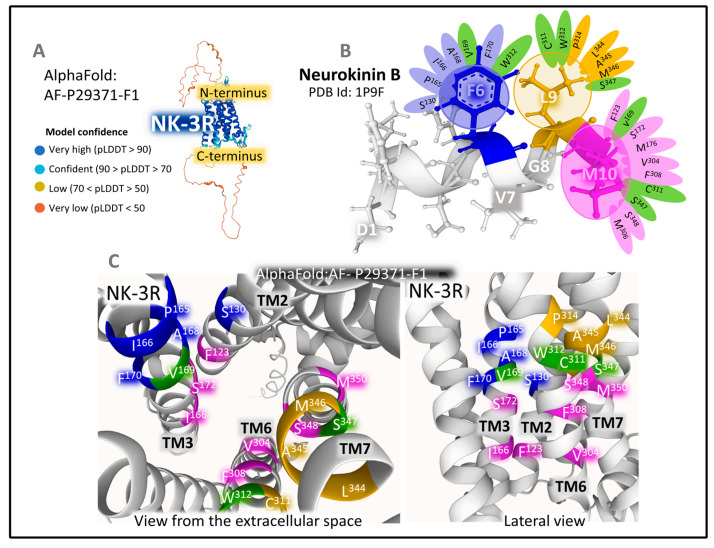
(**A**) represents the predicted structure of NK-2R (UNIPROT P29371) [25] according to the prediction Alpha-Fold model [62,63,64]. The parameter pLDDT (predicted local distance difference test) assessed the model confidence. (**B**) depicts the 3D structure of NKB determined by NMR obtained from the Protein Data Bank [65] (PDB ID 1P9F, [67]) and drawn with Mol* free web-based software [71]. The panel also shows the contacts of NKB’s C-terminal amino acids with the receptors’ amino acids. (**C**) illustrates two views of a three-dimensional representation of NK-2R, obtained from modeling and docking studies [56], indicating the principal amino-acid positions in the Alpha-Fold model of NK-2R closely contacting the C-terminal region of NKA. Residues in green contact more than one pharmacophore of NKB.

#### 2.1.3. Intracellular Signaling of NK-2R and NK-3R

NKA and NKB favorably bind and activate NK-2R and NK-3R, respectively, but they can also bind and activate NK-1R. Consequently, when studying the intracellular signaling of NK-2R and NK-3R, one should consider the possible activation of NK-1R by both neurokinins. The presence, absence, and abundance of the three neurokinin receptors and their endogenous agonists result in a possible explanation for the multiple outcomes and physiological and pathological consequences in various tissues and organs. Redundance of tachykinin receptors may be a compensatory mechanism [72]. It, however, may also serve excessive stimulation leading to malfunction. For example, NKA commits NK-1R, not NK-2R, in mouse macrophages to activate cellular events dependent on transcription factor NF-κB [72]. Basic information on neurokinin signaling comes from experimental analysis in cell lines and cell expression systems where a total determination of intracellular second messengers points to the activation of specific biochemical cascades. However, the study of signaling endosomes and compartmentalized intracellular signals should also be considered [17]. This section describes the central signaling cascades activated through NK-2R and NK-3R in different tissues. Figure 6 represents how NK-2R and NK-3R recruit the transducers Gαs, Gαq/11, and Gα12/13 protein subunits. As a result, phospholipase C, adenylyl cyclase, and phospholipase A2 turn on, respectively. Downstream reactions include numerous protein kinases that control cellular function through transient phosphorylation of their respective substrates. Therefore, neurokinin signaling results in a pleiotropic display of multiple players that kindle rapid metabolic adaptations and short- and long-term control of gene expression. The effects vary between cells and tissues and depend on the presence and abundance of receptors. The concentrations and the agonists’ availability also affect the receptor protein’s functionality. Deregulation of these signaling systems causes severe alterations in cell proliferation, cell migration, and inflammation.

Briefly, we describe representative signaling pathways susceptible to activation by NK-2R and NK-3R that may play a role in cell malfunction leading to cancer development.

Activation of adenylyl cyclase generates cAMP, which triggers protein kinase A (PKA) [73,74]. The nucleotide cAMP, independently of PKA, also activates Epac (guanosine exchange proteins directly activated by cAMP) and nonselective cation channels [75]. Numerous substrates of PKA include phosphorylase kinase, GSK3 (glycogen synthase kinase), CaMKII (calcium–calmodulin kinase), and the transcription factor CREB (cAMP-responsive element binding protein), to mention a few [74]. The aberrant function of the signaling axis cAMP/PKA/CREB and the abnormal behavior of other PKA substrates may cause altered cell metabolism and gene expression affecting cell growth, proliferation, migration, and cell adhesion, driving the development of tumors in different tissues [76].

The PLC/IP3/intracellular calcium axis activates PKC (protein kinase C) and other calcium-dependent kinases: CaMKII or PI3K (phosphatidylinositol 3-kinase). PKC is a serine–threonine kinase with copious potential substrates, including, for example, GSK3, histones, integrins, dynamins, the apoptosis regulator Bcl2 (B-cell lymphoma), Bad (Bcl-associated death protein), and the proto-oncogene transcription factor cMyc [77]. PKC, therefore, is a crucial enzyme with a regulatory role associated with many aspects of cell survival and proliferation.

Heterotrimeric Gα12/13 proteins regulate signaling pathways that modulate cell functioning through several targets. One primary target, through Rho GTPase, is the protein kinase ROCK (Rho-associated coiled-coil kinase) [78]. ROCK modulates cytoskeleton organization and assembly by controlling the activity of LIMK (LIM protein domains kinase) and MLC (myosin light chain) phosphatase. In endothelial cells, ROCK regulates angiogenesis by activating transcription factors, such as c-Fos, c-Jun, or HIF-1α (hypoxia-inducible factor) [79]. ROCK proteins also maintain cell proliferation by directing cell-cycle cyclins (cyclin A) and cyclin-dependent kinases (CDK1) [80]. ROCK phosphorylation is associated with the induction of cancer stem cells, cell invasion and metastasis, angiogenesis, dysregulated energy metabolism, and altered cell proliferation. Therefore, its participation in cancer deserves attention [79].

The hydrolytic activity of phospholipase A2 (PLA2) (EC 3.1.1.4) on membrane phospholipids generates two products: mainly arachidonic acid (AA), esterified to the sn-2 position of glycerol, and lysophospholipids [81]. Lysophospholipids regulate cell function by activating specific GPCRs and other effector proteins [82]. The other product of the reaction catalyzed by PLA2 is AA. This fatty acid is the metabolic precursor, through different enzyme-catalyzed reactions, of prostaglandins, leukotrienes, eicosanoids, and prostacyclins. Metabolic products of AA are involved in multiple cellular responses responsible for inflammation, cell survival, cell proliferation, and cell migration [83].

The activation of NK receptors by neurokinins depends on several factors, from receptor number, activated states, receptor occupancy, recruited effectors, and allosteric regulators to neurokinin concentrations and their competitivity for the three receptors NK-1R, NK-2R, and NK-3R. The intricacy of signaling pathways and their fine-tuned regulation protect the cell by employing redundant reactions and controls. Cancer development requires the malfunction of several biochemical mechanisms that make it difficult for the cell to overcome. Not only can all activate signaling pathways triggered by tachykinins crosstalk, but also those dependent on the activity of agonists acting on other co-existing GPCR receptors [84].

#### 2.1.4. The Expression of NK-2R and NK-3R

The NK receptors appear in the central and peripheral nervous system, endothelial cells, smooth muscle, or blood cells (lymphocytes, neutrophils, and macrophages), where they control multiple biochemical mechanisms attending endocrine, inflammation, or smooth muscle activity regulation [85,86].

According to the human protein atlas [87], the expression of NK-2R, measured as mRNA, distributes in different tissues, but the higher presence appears in the gastrointestinal tract, kidney, urinary bladder, muscle tissues (smooth, heart, and skeletal), monocytes, and female and male tissues. Its detection in the brain and other tissues is scarce. The mRNA coding for the NK-3R is detected in the brain, eye, kidney, and urinary bladder with a weak signal in other tissues (lung, pancreas, or gastrointestinal tract). The mRNA coding for the NK-2R increased in human cancer samples from the gastrointestinal tract, testis, breast, endometrial, cervical, and urothelial tissues. The upregulation of mRNA coding for the NK-3R was higher in human lung, breast, urothelial, endometrium, testis, and ovarian cancer than in normal tissues [17,88,89].

Alterations in the expression profiles of *TACR2* and *TACR3* genes have been reported for different pathologies, e.g., polycystic ovarian syndrome [90], leiomyoma [91], oral carcinoma [21], or breast cancer [92]. However, further studies will reveal why, how, and to what extent dysregulation of NK-2R and NK-3R affects cell function and may provoke abnormal cell growth and development. The altered activity of NK-2R and NK-3R has to be considered together with the modified action of NK-1R.

### 2.2. Calcitonin/Calcitonin Gene-Related Peptide Family

The peptides of this family (adrenomedullin, adrenomedullin 2, amylin, and calcitonin gene-related peptide) regulate the secretion of hormones and are widely distributed by the body [93,94]. Many tumor types have reported AM expression and secretion [95]. AM exerts an antiapoptotic effect in endothelial cells, and cancer cells promote angiogenesis, regulate the permeability of endothelial cells, and contribute to the differentiation of bone marrow-derived mononuclear cells into endothelial progenitor cells. It is also involved in several pathologies [95,96,97,98,99,100,101,102,103,104,105,106]. Amylin, an islet amyloid polypeptide, regulates insulin secretion/glucose homeostasis and is a crucial constituent of the amyloid in insulinomas [107,108,109,110]. Widespread distribution of calcitonin gene-related peptide (CGRP) throughout the peripheral and central nervous systems has been reported; CGRP has been located in myelinated A gamma fibers, small-diameter sensory C fibers, and unmyelinated fibers [93].

#### 2.2.1. Genes and Products of Human Calcitonin

The human calcitonin family of peptides includes calcitonin (CT), α and β forms of CGRP, amylin (AMY or islet amyloid polypeptide (IAPP)), AM, and AM2 or intermedin. Their amino-acid sequence length varies from 32 in CT to 52 for ADM [47] (Figure 7). Although the amino-acid sequence among the six peptides is variable, they share common architectural traits essential for functionality, including a coil structure, a central area showing a helical or disordered structure, and an amidated C-terminal amino acid [111,112,113]. These peptides activate a family of receptors belonging to class B GPCR to serve numerous and diversified functions, such as plasma calcium reduction (CT), food intake modulation (AMY), and vasodilatation and inflammation (CGRPs) in different tissues [112,113,114].

Five human genes encode the calcitonin/CGRP family of peptides. CALCA, CALCB, and AM are on chromosome 11, IAPP is on chromosome 12, and AM2 is on chromosome 22 [34,113]. The peptide products synthesized are tissue-specific in physiological and pathological situations because spliced variants transcripts encode different products. For example, a *CALCA* gene transcript variant encodes calcitonin in the thyroid gland in physiological conditions. Another variant of the *CALCA* gene encodes the neuropeptide CGRPα in the neural tissues [116] (Figure 8).

All six genes encode precursor peptides (pro-peptides) that undergo post-translational modifications (hydrolysis on a pair of aromatic residues, disulfide bridge formation at the N-terminal region, and amidation of the C-terminal amino acid) to attain a functional structure. The *CALCA* gene encodes CT and the α form of CGRP, and the *CALCB* gene encodes the β form of CGRP. The *IAPP*, *AM*, and *AM2* genes encode AMY, AM, and AM2/intermedin peptides [113,114,116,117]. In addition to the peptides enumerated, this family of genes generates other peptides with biological activity. One of them is katacalcin [118], a calcium-lowering hormone that occupies the pro-calcitonin I precursor’s amino acids from 121 to 141.

#### 2.2.2. Structure and Dynamics of the CGRP Receptor: A Three-Component Complex

The structural states of CGRP receptors (CGRPR), in both their apo and their ligand-bound forms, are essential to understand how structural dynamics direct the recruitment of intracellular proteins responsible for signaling mechanisms leading to molecular changes and cellular adaptations [119]. Defining the functional architecture of protein receptors requires fine studies at the atomic level that unveil how ligands (agonists and antagonists) interact with the protein. Combining CLR (calcitonin receptor-like receptor) and RAMPs homologous (receptor activity modifying protein 1) 1, 2, and 3 generates different receptors with variable affinities for endogenous ligands. The CGRPR (calcitonin-gene-related peptide receptor) is a heterodimer formed by CLR and RAMP1 exhibiting a high affinity for CGRP. CLR-RAMP2 (AM-1-receptor) and CLR-RAMP3 (AM-2-receptor) bind with higher affinity to AM, and CTR (calcitonin receptor) associated with RAMP1 is the AMY-1-receptor [120,121]. CLR belongs to class B1 (secretin), subfamily CT-like [122] of GPCR (G-protein-coupled receptors). The human CLR protein (UNIPROT Q16602) comprises 461 amino acids and exhibits post-translational modifications affecting the amino- and carboxy-terminal ends (Figure 9).

RAMP1 is a chaperone protein with a single transmembrane domain that allosterically modulates CLR function [123]. It sits within transmembrane (TM) helices TM3, TM4, and TM5 of CLR (Figure 10). The signaling efficiency of the receptor complex CLR-RAMP activated with different affinities by endogenous ligands CGRP, AM, AMY, and CT improves with a third proteinaceous component directly interacting with CLR, the RCP (receptor component protein) [124,125,126].

A crystal structure of the ectodomain of the CGRP receptor bound to the selective antagonist telcagepant revealed the interactions between the antagonist compound and the lateral chains of amino acids from RAMP1 and CLR (Figure 10C,D) [127]. Telcagepant and olcegepant exert their antagonistic effect by disrupting the access of the endogenous peptide agonist to the binding site located in the interphase between CLR and RAMP1 [127].

Structural studies also revealed the binding of antibodies to halt CGRPR action. Erenubab is a monoclonal IgG antibody approved by the US FDA (Federal Drug Administration) for the treatment of migraine. The antibody is highly selective and potent, and it exhibits full antagonism for the CGRPR [128]. X-ray crystallography and functional studies [129] determined the interaction of erenumab with the receptor complex (Figure 11). Several regions of the light and heavy chain of the antibody, named CDRs (complementary determining regions), are in close contact with the interface between CLR and RAMP1. Given its conformational plasticity, region CDRH3 in the heavy chain, including residues 99 to 119, represents a key sequence responsible for receptor recognition and selectivity. Eventually, the interaction of the antibody with the receptor complex impedes the access of the agonist to its binding site and, consequently, exerts its antagonist effect [129].

Distinctive coupling of antibodies and small non-peptide antagonists to the receptors may explain differences in selectivity, efficacy, and potency with important consequences in designing molecules that control the CGRP receptor’s activity [130].

#### 2.2.3. Mechanisms of Signaling of the CGRP Receptor

The heterodimer CLR-RAMP1 shapes the fully functional CGRPR. RAMP1 serves several functions, from determining, together with CLR, the peptide binding site to helping the glycosylation of the N-terminal segment of CLR and the intracellular trafficking of the receptor [131,132]. However, the high-affinity state of the receptor requires the recruitment of RCP, a peripheral membrane protein, acting as an allosteric modulator that augments the effective functioning of the receptor complex but does not modify agonist binding [126]. In cell cultures, eliminating RCP by knockout significantly reduced cAMP production by stimulating CGRP receptors [126]. Interestingly, this protein does not favor the recruitment of Gq protein, does not participate in Gq-conveyed events [133], and may contribute to CGRPR bias signaling. The precise mechanism explaining the influence of RCP on CGRP receptor signaling is not fully understood [132].

Like many other GPCRs, the CGRP receptor signals through several kinase cascades, which amplify the signal to modulate many intracellular components, from metabolic enzymes to transcription factors [38,84]. As CGRP may signal through CGRP receptors, AM, and AMY receptors, delimiting the activity of CGRP only through CGRP receptors may be difficult [132]. However, signaling profiles depend on the selectivity of the agonist-receptor binding, the availability of signal transducers, and intracellular kinase targets.

Figure 12 summarizes the principal routes of intracellular signaling activated by the heterodimer CLR-RAMP1 with the allosteric aid of RCP. CGRP receptors recruit Gαs subunits to activate adenylyl cyclase and set PKA in action. Various PKA substrates (from K^+^ channels to transcription factors and enzymes) amplify and diversify the primary signal [134,135,136,137] CGRP receptors may also trigger Gαq subunits, leading to PKC activation.

However, CGRP receptors do not only signal from the plasma membrane settlement. Upon activation, the CLR component suffers phosphorylation and β-arrestin recruitment leading to internalization [132]. A clathrin and dynamin-dependent endocytosis can transport the receptor complex to the cell’s interior. Once in the cytoplasm, the complex may locally signal to regulate different pathophysiological outcomes [138].

It is necessary to consider that most experiments and data concerning CGRP receptor signaling come from studies on cell cultures or cells transfected with the receptor complex. In vivo studies are necessary to corroborate the receptor’s role in a more physiological context.

## 3. Involvement of the Tachykinin and Calcitonin/Calcitonin Gene-Related Peptide Families in Cancer

### 3.1. Tachykinin Peptide Family

SP is the most studied tachykinin, including its implication in cancer [1,5]. However, other main tachykinins are less studied; thus, the participation of both NKA and NKB in cancer is reviewed (Table 1).

#### 3.1.1. Breast Cancer

NKA exerts a proliferative action on breast carcinoma tumor cells expressing NK-2R [139]. Compared to non-metastatic breast cancer cells, overexpression of NK-2R/NK-1R was reported in metastatic breast cancer cell lines [140]. Malignant breast biopsies and breast cancer cells lines showed an increased expression of both NK-1R and pre-protachykinin A compared with benign breast biopsies and normal mammary epithelial cells; the level of NK-2R was high in both malignant and normal cells [92]. NK-2R and NK-1R antagonists blocked the proliferation of breast cancer cells, and this suggests that an autocrine stimulation of these cells occurs via pre-protachykinin A peptides (NKA, SP) [92]. Importantly, NK-2R mediated the proliferation of breast cancer cells but not that of normal cells [92].

NKA and SP increased the aggressiveness of a metastatic breast cancer cell line by increasing its ability to migrate and invade tissues [141]. Both peptides augmented the expression of NK-2R and NK-1R on the metastatic breast cancer cell line and, in addition, promoted the release from these cells of the high-molecular-weight kininogen molecule (bradykinin precursor) which exerted tumorigenic and pro-nociceptive actions [141]. Significantly, SP only increased the expression of the NK-1R, but NKA increased the expression of both NK-2R and NK-1R in the mentioned breast cancer cell line [141].

#### 3.1.2. Colorectal Cancer

A high expression of the *NK-2R* gene has been related to poor survival in patients with colorectal cancer; this expression was increased by interferon-α/β in a Janus kinase 1/2-dependent manner [143]. NK-2R rs4644560 GC polymorphism alone or in combination with NK-1R rs10198644 GC is a prognostic marker for lymph node metastasis in patients with colorectal cancer [142]. Patients with NK-2R rs4644560 CC showed lesser positive lymph nodes than those with rs4644560 GC, and the number of positive lymph nodes was increased in NK-2R rs4644560 GC/NK-1R rs10198644 GG patients compared to NK-2R rs4644560 GG/NK-1R rs10198644 GG individuals.

Colorectal tumor cells overexpressing NK-2R showed increased tumorigenesis and metastatic colonization in vivo experiments; this expression has been associated with the malignancy of colon cancer cells [143]. SP reduced the invasive potential of colon carcinoma cells, but NKA had no effect [167]. NKA increased the viability and proliferation of colon cancer cells and the phosphorylation of extracellular-signal-regulated kinase 1/2 levels in interferon-α/β-treated colon cancer cells, whereas NK-2R antagonists reduced the proliferation of these cells [143]. However, NKA did not exert growth-regulatory actions on a human colon cancer cell line (HT 29) and a non-transformed small-intestinal cell line from the rat (IEC-6) [168].

#### 3.1.3. Glioma

NKA promoted the proliferation and release of interleukin-6 from glioma cells expressing NK-1R; these actions mediated by NKA were entirely blocked with a specific NK-1R antagonist (MEN-11467) [144]. The above means that NKA exerted its actions not only via NK-2R but also through NK-1R.

SP binding sites but not NKA binding sites have been reported in an astrocytoma cell line (U 373) [169]. NKA and NKB promoted taurine release from astrocytoma cells. NKA induced a greater release than NKB; however, SP promoted the most significant release of taurine from these cells [170]. Astrocytoma cells express NK-3R, and endocytosis of copper-bound NKB occurs through a trafficking pathway that includes early endosomes [145].

#### 3.1.4. Insulinoma

RIN5mF cells (rat insulinoma cells) express the *pre-protachykinin A* gene and release NKA and SP [146,147]. No change in the concentration of NKA/SP was observed in the small intestine and stomach of insulinoma rats [171].

#### 3.1.5. Lung Cancer

Pulmonary carcinoid tumors express mRNA pre-protachykinin A; in some, NKA/SP has been reported [148]. NKA and NKB inhibited the growth of small-cell lung cancer cells [20].

#### 3.1.6. Medullary Thyroid Carcinoma

NKA has not been detected in medullary thyroid carcinoma [172].

#### 3.1.7. Midgut Carcinoid Tumor

Pre-pro-tachykinin A mRNA expression and the presence of NKA/SP have been reported in midgut carcinoid tumors and in the plasma of patients with midgut carcinoids [148,149]. These tumors show an unpredictable clinical behavior; hence, specific biomarkers are needed to detect the disease early [150]. Plasma NKA level is an excellent biomarker of prognosis; patients with a high NKA level showed worse survival than those with a decreased or stabilized level [150]. Notably, the most recent plasma NKA level predicted a better survival than the initial value of such a level. Moreover, increased luminal content of tachykinins (e.g., NKA and NKB) has been reported in malignant midgut carcinoids compared with that reported in healthy individuals [151].

#### 3.1.8. Neuroblastoma

NKB but not NKA/SP has been located in neuroblastoma [152]. NK-2R and NK-1R mediate the proliferation exerted by pre-protachykinin A peptides on neuroblastoma cells [153]. The murine neuroblastoma C1300 cell line expresses mRNA NK-2R/NK-3R, but not mRNA NK-1R and NKA; NK-3R increased its cytosolic Ca^++^ concentration, which was inhibited with the NK-2R antagonist SR-48968 [154,155]. NK-2R and NK-3R are activated independently by NKA; the activation of both receptors promoted not only a Ca^++^ increase but also the formation of inositol trisphosphate, whereas both mechanisms were blocked with phospholipase C inhibitors [154]. The results demonstrated that NK-2R/NK-3R-mediated Ca^++^ increase was due to the activation of phospholipase C and, in addition, it was dependent on the entry of extracellular Ca^++^ via voltage-independent channels and the mobilization of internal Ca^++^ stores [154].

#### 3.1.9. Oral Squamous Cell Carcinoma

NK-3R expression was very high in oral squamous tumor cells, whereas, in normal epithelial cells, NK-3R was not observed [21]. Moreover, those squamous cells that invaded the mandible bone matrix showed a higher expression of NK-3R and using the selective NK-3R antagonist SB-222,200 significantly inhibited tumorigenesis and the osteolytic lesion [21,156]. Cancer cells did not express NKB, but this peptide was observed in sensory nerves in the mandible. This discovery suggests that the release of NKB from these nerves could regulate the proliferation of tumor cells.

#### 3.1.10. Phaeochromocytoma

NKB has only been detected in one of the 10 phaeochromocytomas studied; in the same study, NKB was not detected in carcinoid tumors [157]. Moreover, in a human phaeochromocytoma extract, NKA/SP was detected [31].

#### 3.1.11. Schwannoma-Derived Cells

NKA and SP were located in the cytoplasm of malignant schwannoma-derived cells [158]. Both peptides seem to play a role in the tumor microenvironment; however, future studies must elucidate this.

#### 3.1.12. Small Bowel Neuroendocrine Tumors

Small bowel neuroendocrine tumors are difficult to diagnose; thus, it is important to determine specific biomarkers associated with the disease. NKA is a specific blood biomarker because high plasma NKA levels (≥50 ng/L) have been associated with poor prognosis in patients with these tumors. Monitoring such levels could be useful in selecting patients with a poor prognosis; hence, a high NKA level means that an urgent therapeutic intervention is needed [159,160,161,162,163]. Thus, plasma NKA levels predict survival in patients with small bowel neuroendocrine tumors. An improved prognosis was reported by lowering the plasma NKA level below 50 ng/L [161,163].

Compared with healthy individuals, a rise in circulating NKA/SP has been reported in patients suffering from ileal metastatic carcinoid tumors showing cutaneous flushing; the release of both peptides from carcinoid tumors was partially blocked after the administration of a somatostatin analog [164]. The presence of NKA, NKA_3-10_, and NKA_4-10_ has been detected in ileal metastatic carcinoid tumors [165]. Therefore, NKA shows an N-terminal heterogeneity in these tumors, and carcinoid tumors can release different amounts of several tachykinins contributing to individual differences.

#### 3.1.13. Uterine Leiomyomata

Leiomyoma, a benign smooth muscle tumor, showed an upregulation of NK-2R mRNA compared with normal myometrium; however, the levels of NK-2R protein were similar in both tumor and normal cells [166]. Moreover, leiomyomas express NKB and NK-3R, which were significantly more highly expressed in this benign tumor than in normal myometrium [91]. NKB was observed in the nuclei of smooth muscle cells in normal myometrium, whereas leiomyoma cells showed a predominant cytoplasmic expression of the peptide [91]. Estrogens control NK-3R, and the activation of this receptor promoted nuclear translocation affecting gene expression and chromatin structure [91]. These results suggest that the NKB/NK-3R system is involved in the pathological events observed in women suffering from leiomyomata.

Figure 13 shows the main findings mentioned in this section regarding the implication of NKA/NKB in cancer.

### 3.2. Calcitonin/Calcitonin Gene-Related Peptide Family

Extensive data have shown the involvement of this peptide family in cancer. For example, adrenomedullin (AM) is released from choroid plexus carcinoma [173]; AM released from tumor cells drives both tumor and lymph node lymphangiogenesis, and AM gene dosage has been related to both mechanisms [174]. Tumor cells express/overexpress AM (e.g., ovary, prostate, kidney, skin, endometrial, liver, pancreatic, brain, and breast cancer), and the level of AM has been correlated with cancer severity [104,175].

AM acts as a mitogenic agent on tumor cells and favors a more aggressive tumor phenotype. Under hypoxic conditions that occur in the proximity of solid tumors, the peptide is upregulated via an HIF 1-dependent pathway in normal and tumor cells, promoting angiogenesis [176,177,178,179]. AM also prevents apoptosis, suppresses the immune system, and is involved in bone metastasis [180]. Adrenomedullin 2 (AM2/intermedin) and CGRP act as tumor survival/growth factors promoting lymphangiogenesis and angiogenesis [175]. Lastly, the infection of dermal endothelial cells with live and UV-inactivated Kaposi’s sarcoma-associated herpesvirus (KSHV) demonstrated that the viral gene expression was responsible for the upregulation of the *AM2* gene [181]. The implication of AM, AM2, AMY, and CGRP in 29 tumors is reviewed in Table 2.

#### 3.2.1. Acute Myeloid Leukemia

##### Adrenomedullin

AM, CLR, and RAMP2 and 3 have been reported in acute myeloid leukemia. A high level of AM has been associated with low overall survival and disease-free survival [182,183]. AM and AM_22–52_ fragment regulate the growth of leukemia cells through autocrine and paracrine mechanisms [184]. The expression of AM in acute myeloid leukemia has been associated with genes related to immunosuppression and a stem cell phenotype, leading to disease relapse and therapy resistance [185]. This observation is vital since drug-tolerant and -resistant leukemia stem cells have been associated with relapses in acute myeloid leukemia. AM via CLR has been correlated with adverse outcomes of acute myeloid leukemia [183]. Depleting this receptor decreased leukemia stem cell frequency of relapse-initiating cells post chemotherapy in vivo. CLR knockdown also decreased leukemia stem cell frequency and impaired leukemia cell growth [183]. This finding suggests that AM is involved in the post-chemotherapy persistence of these cells, and that targeting CLR could prevent relapse in acute myeloid leukemia [183].

##### Calcitonin Gene-Related Peptide

A possible origin of leukemogenesis is the disruption of DNA methylation patterns, and, in acute leukemias, the *CT* gene plays a crucial role in gene hypermethylation [186]. The *CT* gene methylation pattern is an independent prognostic factor in pediatric acute leukemia that could characterize a group of patients with enhanced risk of relapse and death [186]. This gene was found to be hypermethylated in children with acute myeloblastic leukemia (54.3%) or with acute lymphoblastic leukemia (65.7%) [186]. This hypermethylation was not associated with response to induction therapy, standard prognostic factors, and clinicopathologic characteristics.

A high CT receptor expression has been related to a poor prognosis in acute myeloid leukemia. This receptor was upregulated in leukemic stem cells and at relapse of acute myeloid leukemia [182]. Moreover, CT receptor expression was correlated with chemotherapy resistance [182]. In experimental animal models of acute myeloid leukemia, olcegepant (CGRP antagonist) increased cell differentiation and decreased leukemic burden and stem-cell characteristics [182]. Moreover, MK0974 (CGRP antagonist) blocked the CLR/RAMP1 complex and promoted apoptosis in the EVI1 acute myeloid leukemia cell line, and this antagonist also attenuated p38 and ERK phosphorylation in this cell line [136]. Thus, blocking the CT receptor/CGRP system could be an antitumor strategy to treat this disease.

#### 3.2.2. Adrenocortical Tumor

##### Adrenomedullin

AM is synthesized and released from adrenocortical tumors and phaeochromocytomas [189], with the release of AM controlled by cytokines [286]. It has been suggested that the low/undetectable level of AM immunoreactivity in adrenocortical tumors is due to a rapid release of AM from tumor cells [189]. mRNA expression of AM and its receptor RAMP2/CLR has been reported in phaeochromocytoma tissues and the normal adrenal medulla. Nevertheless, this expression was higher in phaeochromocytomas than in normal medulla [190]. The RDC1 receptor (an AM putative receptor) is expressed differently in benign and malignant phaeochromocytomas. The authors suggested that AM/RDC1 is involved in chromaffin cell tumorigenesis through pro-survival actions [287].

AM plays a part in the pathogenesis of phaeochromocytoma; the peptide is increased in this disease, and its level has been correlated with plasma noradrenaline levels [288,289]. AM blocked the proliferation of human phaeochromocytoma cells [190], and the level of AM has been suggested to be a biomarker in patients with phaeochromocytoma [191]. Circulating AM levels increased in patients with phaeochromocytoma compared with patients with nonfunctional adrenocortical adenomas and healthy individuals [289]. Moreover, plasma chromogranin levels correlated with plasma AM/metanephrine concentrations in phaeochromocytoma patients [289]. AM expression was controlled by the nerve growth factor in phaeochromocytoma PC12 cells, but its expression was inhibited during the neuronal differentiation of these cells [290].

##### Adrenomedullin 2

Phaeochromocytomas and aldosterone-secreting adenomas express AM2 [187]. This expression has been reported in human adrenal tumors of both medullary or cortical origins and in attached non-neoplastic adrenal tissues [192]. However, this expression was lower in the non-neoplastic portion of attached adrenal cortices than in adrenocortical tumor cells [192]. Moreover, AM2, CLR, RAMP1, RAMP2, and RAMP3 mRNA expressions were also observed in both adrenal tumors and attached adrenal tissues [192]. Thus, it seems that AM2 is involved in tumor growth.

##### Calcitonin Gene-Related Peptide

A high CGRP level has been reported in tissues from phaeochromocytomas, and plasma CGRP level was slightly increased. Nevertheless, it did not change significantly with tumor manipulation or early after tumor resection [188].

#### 3.2.3. Bladder Cancer

##### Adrenomedullin

Patients with bladder urothelial cell carcinoma showed a higher presence of AM in the tumor than in adjacent nontumor bladder regions. Under hypoxic conditions, AM expression increased in human bladder cancer cell lines [193]. Moreover, compared to the control group, AM knockdown in T24 cells promoted apoptosis, and the combination of AM knockdown and cisplatin decreased tumor growth when compared with cisplatin treatment alone or AM knockdown alone [193]. This finding indicates that AM is involved in the development of bladder cancer, and that AM may represent a potential antitumor target.

#### 3.2.4. Breast Cancer

##### Adrenomedullin

Overall, 82% of the breast cancer samples studied showed a moderate to strong staining for AM, and this expression was associated with axillary lymph node metastasis [194]. AM plasma level reflects the primary tumor size in breast cancer patients. Nevertheless, no difference was reported between the AM plasma levels of healthy individuals and patients with breast cancer [194]. Thus, the level of circulating AM cannot be used as a tumor marker in breast cancer, but it could be a predictor for lymph node metastasis.

Human breast cancer cell lines (T47D and MCF7) express a low level of AM; however, when they were transfected with an expression AM construct, both cell types expressed a high level of AM (protein and AM mRNA) [99]. T47D and MCF7 cells overexpressing AM increased their angiogenic potential and showed less apoptotic mechanisms after serum deprivation and a more pleiotropic morphology [99]. Both cell types did not show motility, but ECV ovarian tumor cells treated with AM showed higher motility than saline-treated ECV cancer cells [99]. Moreover, compared with control cells (T47D cells transfected with empty vector), T47D cancer cells overexpressing AM showed a higher level of proteins involved in oncogenic signal pathways (e.g., mitogen-activated protein kinase (MAPK) p49, protein kinase C, Raf, and Ras) and a lower level of proapoptotic proteins (e.g., caspase 8, Bid, and Bax) [99]. Importantly, animals (three of 10) xenografted with T47D cells overexpressing AM developed tumors, whereas none of the animals xenografted with cells carrying the empty plasmid developed tumors [99]. The data show that AM is a tumor survival factor and that the blockade of AM could be a promising antitumor strategy.

It is known that the lysyl oxidase (LOX) L2 enzyme plays an important role in tumor expression since it induces the epithelial-to-mesenchymal transition (EMT) in cells, which is an important mechanism for tumors to become metastatic. In the MDA-MB-231 metastatic breast cancer cell line, a link between RAMP3 and LOXL2 has been reported; the blockade of LOXL2 promoted a mesenchymal-to-epithelial transition in breast cancer cells, a decreased invasive phenotype, and RAMP3 expression, leading to a reduced tumor development and tumor microvessel density when compared with controls [291]. Therefore, RAMP3 is involved in cancer metastasis. Breast tumor cells, expressing and releasing AM, favor cell proliferation, breast cancer bone metastasis, and angiogenesis, as well as stimulate bone formation and osteoblast activity [195]. This discovery suggests that the peptide is involved in skeletal metastases. Significantly, AM antagonists blocked bone tumor growth and decreased the markers for osteoclast activity [195]. AM expression diminished in triple-negative breast cancer samples and cell lines. This low expression has been related to poor prognosis and an increased risk of tumor recurrence and metastasis [196]. AM could be used as a biomarker for triple-negative breast cancer prognosis. The peptide could act as an antimetastatic agent in this disease because AM, via its effect on cancer cell EMT, decreased tumor cell invasion [196].

Tumor cells interact with cells located in their environment to favor tumor growth and invasion. Cancer-associated fibroblasts are involved in tumorigenesis and angiogenesis. The importance played by the cancer-associated fibroblast-derived AM system in neovascularization and breast carcinoma growth has been demonstrated [197]. AM_22–52_ treatment disrupted the vasculature of tumors, depleted vascular endothelial cells, decreased tumor cell proliferation, and induced apoptosis [197]. Moreover, breast cancer cells release AM, which regulates the activity of cancer-associated adipocytes, promoting delipidation and metabolic changes [292].

##### Adrenomedullin 2

Pre-operative plasma AM2 level has been related to poor outcomes in breast cancer patients; hence, it can be used as a prognostic biomarker for these patients [198]. AM2 expression is increased in breast cancer samples. Its level was positively correlated with both Ki67 expression and lymph node metastasis and promoted the growth, migration, and invasion of breast cancer cells; all these actions were blocked with a monoclonal anti-AM2 antibody [199]. AM2 increased the malignancy of breast cancer cells, upregulated the expression of ribosomal component genes by activating the Src/c-Myc signaling pathway, improved tumor blood perfusion, and was involved in vascular remodeling [199].

##### Calcitonin Gene-Related Peptide

Neoangiogenesis and CGRP expression are increased in mixed invasive–preinvasive breast lesions [200], and CGRP is involved in breast cancer metastasis [201]. CGRP regulates osteoclast coupling genes in the MG-63 osteoblast cell line by decreasing RANKL/Runx2 expressions, increasing OPG expression, and blocking the osteolytic factors induced by the interaction between osteoblasts and breast cancer cells [201]. This blockade was reverted with a CGRP antagonist.

#### 3.2.5. Choriocarcinoma

##### Adrenomedullin

AM is synthesized and released from cytotrophoblastic cells expressing the AM receptor [202]. AM mRNA expression was reported in human placental trophoblastic tissues and choriocarcinoma JAr cells and AM receptors in trophoblastic cells [202].

#### 3.2.6. Colon Cancer

##### Adrenomedullin

CLR, RAMP2, and RAMP3 expressions have been reported in colorectal cancer; high levels of AM, CLR, RAMP2, and RAMP3 correlate with lymph nodes and distant metastasis, and a high level of AM correlates with a low disease-free survival [203,204]. AM concentration is higher in colorectal cancer tissues than in adjacent normal tissues, and the presence of AM has been reported in inflammatory bowel disease-derived colorectal cancer [179,205]. A higher AM mRNA expression was observed in patients with colorectal cancer (clinical stages I, III, and IV) than in healthy individuals [204]. AM expression has been associated with vascular endothelial growth factor (VEGF) and HIF-1α in colorectal cancer; these molecules are involved in angiogenesis, and the authors of the study suggested that the degree of AM expression could be used as a marker for the prediction of cancer-related death and high risk of relapse in colorectal cancer patients with a curative resection [203]. DLD-1, a human colorectal carcinoma cell line, increased AM immunoreactivity and the level of mRNA AM under hypoxic conditions; this was also found when these cells were treated with cobalt chloride, a compound that mimics hypoxic states [293]. The conclusion is that AM is an important agent in ischemic states.

HT-29 human colon carcinoma cells synthesized and released AM, and this effect increased under hypoxic conditions. Treating these cells with AM promoted cell proliferation and invasion [204]. By activating the phosphatidylinositol 3-kinase/Akt pathway or upregulating B-cell lymphoma (Bcl)-2, AM exerts an antiapoptotic effect. The AM expression level has been associated with clinical survival rate and cancer stage in colon cancer [206]. This level was higher in colon cancer than in normal tissues, and a relation between AM expression and clinical or pathological parameters was not found in stomach cancers [206]. AM is an upregulated gene in the colon cancer cell line DKs5, which expresses the KRAS oncogene (involved in metastasis, angiogenesis, and chemoresistance) under hypoxia [207]. Knockdown of AM in colon tumor xenografts promoted apoptosis and inhibited angiogenesis, leading to the suppression of tumors [207]. Thus, AM is involved in colon cancer development. A mouse model of colon cancer treatment with the AM positive modulator, 145425, showed a lower number of tumors when compared to the control; this modulator controlled the expression of the Lgr5 proliferation marker [208].

##### Adrenomedullin 2

Colorectal adenocarcinomas showed a higher expression of pre-proAM, pre-proAM2, CLR, RAMP2, RAMP3, metalloproteinase (MMP)-9, and VEGF-A mRNAs than the adjacent normal tissues [205]. AM and AM2 were mainly detected in cancer cells, along with MMP-9 in the adjacent stroma [205]. Moreover, a positive correlation between *MMP-9* gene expression and pre-proAM was observed, but not with pre-proAM2 [205].

##### Amylin

AMY is rarely expressed in intestine endocrine tumors [294].

##### Calcitonin Gene-Related Peptide

The number of nerve fibers and neurons containing CGRP was studied in the submucous/myenteric plexuses located in the transitional zone between cancer-invaded areas and morphologically unchanged regions: a decrease in the number of neurons and fibers was observed in both plexuses [295].

#### 3.2.7. Cutaneous Nerve Neuromas

##### Calcitonin Gene-Related Peptide

CGRP release from nerve fibers was observed in saphenous nerve neuromas [209].

#### 3.2.8. Endometrial Cancer

##### Adrenomedullin

Cobalt chloride increased the release of AM in endometrial cancer, favoring angiogenic and tumorigenic activities and the secretion of VEGF from tumor cells [210]. AM, by upregulating the Bcl-2 antiapoptotic protein expression, blocks cell death by hypoxia in endometrial cancer cells [211]. AM is upregulated in endometrial cells by tamoxifen, a nonsteroidal antiestrogen; AM also promotes the growth of carcinoma [101,211]. In the endometrium, the level of AM increased in progression in cells from normal, simple, or complex hyperplasia with or without atypia to grade 1 adenocarcinoma, but Bcl-2 expression decreased [212]. Another study confirmed this observation in which AM expression increased from benign endometrium to endometrial intraepithelial neoplasia and type-1 adenocarcinoma, but Bcl-2 expression decreased in the transition from endometrial intraepithelial neoplasia to carcinoma [296]. Lastly, no correlation between AM, HIF-1α, or Bcl-2 expressions and SUVmax (maximum standardized uptake value) was observed in endometrial cancer [297].

#### 3.2.9. Ewing Sarcoma

##### Calcitonin Gene-Related Peptide

CGRP is expressed in Ewing sarcoma and promotes the proliferation of Ewing sarcoma cells [213]. CGRPβ mRNA is transcribed from the *CT II* gene in humans, and its expression has been reported in Ewing sarcoma [298].

#### 3.2.10. Gastric Cancer

##### Adrenomedullin

BGC-823 gastric cancer cells showed a high expression of AM under hypoxic conditions; the association of AM in angiogenesis and its release under hypoxia in solid tumors have also been reported [299]. AM knockdown expression decreased the levels of B-cell lymphoma 2 and phosphoprotein kinase B and increased the levels of Bcl-2 associated x protein (Bax) and cleaved-caspase 3 [299].

Mast cells observed in solid tumors show different phenotypes in the tumor microenvironment, and patients with gastric cancer show a high mast cell infiltration into the tumor [214]. Tumor-derived AM promoted, via PI3K/Akt signaling pathway, mast cell degranulation, as well as favored tumor cell proliferation and the blockade of apoptosis in gastric cancer cells; these effects were reverted by blocking the release of interleukin-17A from mast cells [214]. These observations indicate that mast cells play an essential role in the development of gastric cancer.

#### 3.2.11. Glioma

##### Adrenomedullin

CLR expression was observed in 95 biopsies of human gliomas of varying grades; astrocytic cancer cells and endothelial cells showed a high immunoreactivity for the receptor [300]. AM receptors were also reported in glioblastoma cells and glioma tissue [217]. AM also exerts an antiapoptotic action in glioma; hence, the blockade of this effect mediated by AM is a promising antitumor strategy [215]. AM has also been suggested as a tumor angiogenic factor in glioblastoma [216]. Exogenously added AM promoted the growth of these cells; the inhibition of AM produced by tumor cells suppressed tumor growth and decreased the density of tumor vessels [217]. It seems then that AM is involved in developing glioblastoma by promoting the mitogenesis of tumor cells and an angiogenic effect. AM mRNA has been associated with tumor type and grade in glioblastoma; a high expression was reported in all glioblastomas studied, but a low expression was observed in anaplastic astrocytomas, with barely detectable AM mRNA levels in oligodendrogliomas and low-grade astrocytomas [218]. The above indicates that AM has a role in the progression of gliomas.

The expression of AM is highly induced during hypoxia in human T98G glioblastoma cells [301]. Moreover, treating these cells with interleukin-1β or interferon γ increased mRNA AM expression and the release of AM in the culture media, whereas tumor necrosis factor α decreased both effects dose-dependently [217,302]. The data show that AM is released from glioblastoma cells and that the peptide is involved in tumor pathophysiology. Dexamethasone increased AM mRNA levels in T98G cells and AM immunoreactive levels in the culture medium but decreased immunoreactive-endothelin-1 levels [303]. Treatment with tumor necrosis factor α, interleukin-1β, and interferon γ promoted the expression of both AM and endothelin-1 in T98G cells [303]. The cAMP level was also increased in glioblastoma cells when they were treated with synthetic AM1-52 [302]; AM favored c-Jun and c-Jun N-terminal kinase (JNK) phosphorylation in glioblastoma cells, and the suppression of c-Jun expression or the inhibition of JNK activation impaired the actions mediated by AM on cyclin D1 and cell proliferation [219]. Thus, the c-Jun/JNK pathway is involved in the growth-regulatory activity mediated by AM in glioblastoma cells.

AM expression was upregulated in temozolomide (TMZ)-resistant glioma samples [221]. miR-1297 targeted AM, blocked its expression, and sensitized glioma cells to TMZ treatment; this could be mediated by the Bax/Bcl-2, Akt, and extracellular signal-regulated protein kinase (ERK)1/2 signaling pathways [221].

The cytokine oncostatin M promoted the expression of AM in astroglioma cells favoring the phosphorylation of activator of transcription-3 (STAT-3), nuclear translocation, and DNA binding to AM promoter. The expression of AM is controlled by STAT-3 in these cells, and AM also increases the migration of astroglioma cells [220]. Therefore, the activation of STAT-3, which is, for example, observed in malignant brain tumors, is crucial for AM expression and glioma invasion and metastasis.

##### Adrenomedullin 2

AM2 expression increased and correlated with higher-grade gliomas [222]. AM2 promotes filopodia formation, increasing the invasive capacity of glioma cells, and the activation of ERK1/2 has been involved in the proliferation, invasion, and malignancy of glioblastoma cells [222]. AM2 improves tumor blood and has also been engaged in hypoxia-induced responses and mitochondrial functions (e.g., regulating the critical components of respiratory complex I) in glioblastoma cells [222].

##### Amylin

AMY promoted the release of inflammatory cytokines (interleukin-6, interleukin-8) from U-373 MG human astrocytoma cells and the production of interleukin-1β in these cells [223].

#### 3.2.12. Head and Neck Squamous Cell Carcinoma

##### Calcitonin Gene-Related Peptide

Head and neck squamous cell carcinoma is highly innervated by peripheral sensory neurons releasing CGRP; the peptide has been involved in oral cancer progression [304]. CGRP secreted from peripheral nerve terminals exerts a paracrine action on oral squamous carcinoma cells, and it has been suggested that, in this disease, CGRP is a bridge target between cancer-associated pain and cancer development because CGRP promotes the algesia transmission to pain centers [225]. CGRP also links perineural invasion and lymph node metastasis in oral squamous cell carcinoma. The pre-operative plasma CGRP level has been suggested to predict lymph node metastasis in this disease [224].

#### 3.2.13. Liver Cancer

##### Adrenomedullin

Hepatocellular carcinoma cells express AM and its receptor and, under hypoxic conditions, favor AM expression [226]; in addition, AM mRNA, HIF-1α and VEGF levels were increased under the same conditions in human hepatocellular carcinoma cell lines [229]. CLR, RAMP2, and RAMP3 have been reported in liver cancer, and a high AM level has been related to increased intrahepatic metastasis [226,227]. AM is upregulated in human intrahepatic cholangiocellular carcinoma tissues (73/133; the second most common type of primary liver cancer) compared with healthy individuals [228]. AM regulates ZEB1 activation, which mediates EMT [228]. This observation means that AM-mediated activation could be a promising antitumor target. AM, through Akt activation, promoted the growth of hepatocellular carcinoma cells, which was inhibited with AM inhibitors [226]. Knockdown of AM expression promoted apoptotic mechanisms in hepatocellular carcinoma cells and, combined with cisplatin, significantly decreased tumor growth when compared with knockdown of AM expression or cisplatin alone [229]. The data show the association of AM with hepatocellular carcinoma development.

Microvessel density and mRNA AM and erythropoietin receptor levels were higher in hepatocellular carcinoma than in nontumor tissues [230]. Both levels correlated with tumor metastasis, pathological differentiation, and capsule invasion in hepatocellular carcinoma [230]. AM expression was also associated with the erythropoietin receptor and microvessel density in the same carcinoma [230]. Thus, AM and erythropoietin receptors may induce angiogenesis in hepatocellular carcinoma. AM is also associated with N-cadherin intensity, vascular invasion, and poor prognosis [226]. Consequently, AM level could be a prognostic factor in hepatocellular carcinoma. Additionally, in patients with this disease, AM mRNA levels were higher in tumor tissues than in adjacent nontumor liver tissues [229].

##### Adrenomedullin 2

A high AM2 mRNA expression was found in human hepatocellular carcinoma tumors, even in the early stage; tumors showed a higher level of AM2 than that observed in adjacent benign liver tissues [231]. A high expression of AM2 has been observed in the mesenchymal area of human hepatocellular carcinoma, with this expression being observed in stromal (e.g., vascular smooth muscle cells, fibroblasts) and endothelial cells [305]. AM2 increased hepatic carcinoma cell proliferation and survival [231] and favored HepG2 hepatic carcinoma cell proliferation; the AM2 receptor antagonist AM217-47 blocked this proliferation [232]. Moreover, the proliferation action mediated by AM2 in these cells was via the activation of the Wnt signaling cascade [232]. AM2 also promoted the proliferation of SMMC7721 hepatic carcinoma cells, as well as upregulated miR-155 expression, and the blockade of miR-155 inhibited the proliferation of SMMC7721 cells mediated by AM2 [306]. The data suggest that AM2 is involved in angiogenesis in human hepatocellular carcinoma, and that the blockade of AM2 activity may represent an antitumor treatment against this carcinoma [233].

##### Amylin

AMY binding sites were observed in the human hepatoblastoma cell line HepG2. The peptide favored adenylate cyclase activity [234].

#### 3.2.14. Lung Cancer

##### Adrenomedullin

Lung cancer cells express CLR and RAMP2/3 [235,307]. AM appeared expressed in most non-small-cell and half of the small-cell lung carcinomas studied [308]. However, AM expression did not correlate with the overall survival of patients, cancer stage, and tumor differentiation. The results reported by the authors do not support the role of AM as an independent survival agent for lung cancer [235]. The aryl hydrocarbon receptor is partially responsible for tobacco-induced carcinogenesis, and it has been reported that AM contributes to the carcinogenicity of tobacco-activated aryl hydrocarbon receptor products [236]. Thus, the AM/aryl hydrocarbon receptor axis seems to play an important role in lung cancer development.

##### Calcitonin Gene-Related Peptide

*CGRP* gene expression has been reported in lung cancer, and patients with small-cell lung carcinomas showed a high serum CGRP, but only 27% showed values above the normal upper range [237,238]. CGRP was observed in a few of the tumors studied; thus, the high serum CGRP level observed was from an extratumoral origin [238]. CGRP expression was similar in normal adult lungs, pulmonary tumorlets, hyperplastic bronchial neuroendocrine cells, and, to some extent, in small-cell lung carcinomas [309]. Moreover, CGRP expression in bronchial carcinoids was similar to that observed in late fetal and neonatal lungs [309].

#### 3.2.15. Melanoma

##### Adrenomedullin

AM, CLR, and RAMP2/3 expressions have been reported in melanoma [239,310] and it has been demonstrated that tumor-associated macrophages, through AM, favor melanoma growth and angiogenesis [239]. The number of tumor-associated macrophages (which express and release AM) in the tumor microenvironment has been correlated with poor prognosis in melanoma [239]. In addition, the secretion of AM by macrophages was upregulated when these cells were cocultured with melanoma cells or with melanoma cells conditioned media [239]. Tumor-associated macrophages favored the migration of endothelial cells, as well as increased B16/F10 tumor growth [239]. AM stimulated melanoma growth and angiogenesis through a paracrine effect (mediated by the endothelial nitric oxide synthase signaling pathway) and an autocrine effect (favoring the polarization of macrophages toward an alternatively activated phenotype) [239]. Human melanomas exhibited higher expressions of AM and AM receptors in tumor-associated macrophages than those found in adjacent normal skins. It is also known that antibodies against AM or AM receptor antagonists decreased tumor-associated macrophages-induced angiogenesis and melanoma growth [239].

#### 3.2.16. Nasopharyngeal Carcinoma

##### Adrenomedullin

It has been mentioned that plasma AM levels could be a biomarker for predicting the long-term prognosis of patients with nasopharyngeal carcinoma [240].

#### 3.2.17. Neuroblastoma

##### Adrenomedullin

Human neuroblastoma SK-N-MC cells express AM receptors, and AM’s N-terminal ring structure is essential for binding to the receptor. AM mRNA expression has been associated with tumor differentiation but not to prognostic markers in neuroblastoma [241,242]. In human IMR-32 and NB69 neuroblastoma cell lines, AM expression appeared augmented under hypoxic conditions, but RAMP2 expression was suppressed under the same conditions [311]. In SK-N-SH human neuroblastoma cells, AM favors, probably via cAMP-protein kinase A-dependent mechanisms, Ca^++^ influx via L-type Ca^++^ channels and ryanodine-sensitive Ca^++^ release from the endoplasmic reticulum; Ca^++^ concentration increase activated neuronal nitric oxide synthase and nitric oxide release [312].

#### 3.2.18. Neuroendocrine Tumors

##### Adrenomedullin

Plasma and tissue AM expressions predict tumor progression in patients with neuroendocrine carcinomas of the gastroenteropancreatic and bronchial systems [243]. High AM plasma level or tumor AM mRNA expression correlates with a worsened prognosis in neuroendocrine, renal, prostate, or ovarian cancer patients [243,313].

#### 3.2.19. Oropharyngeal Squamous Cell Carcinoma

##### Adrenomedullin

It has been reported that Jumonji domain-containing 1A (JMJD1A favors histone demethylation). Moreover, H3K9me1/2 and AM expression predicts progression and prognosis in patients with oral and oropharynx cancers [244]. Moreover, the *JMJD1A* gene targeting AM has been shown to favor tumorigenesis and cell growth [244].

#### 3.2.20. Osteosarcoma

##### Adrenomedullin

AM, CLR, and RAMP2/3 expressions have been reported in osteosarcoma, and a high AM expression has been associated with the degree of metastasis and malignancy [245,246]. A correlation between VEGF and AM expression in the plasma of osteosarcoma patients has been reported. Hypoxia stimulated the expression of both VEGF and AM in MG-63 osteosarcoma cells, whose proliferation decreased when AM was inhibited by AM shRNA [314]. AM was overexpressed in human osteosarcoma tissue, but a low expression was observed in adjacent tissues, and a shallow expression was found in normal tissues [245]. MG-63 cell proliferation increased when AM was added but VEGF inhibition attenuated this action. The decrease in AM blocked CD31 expression, VEGF, and the growth of cancer cells in osteosarcoma experimental animal models [314]. Thus, it seems that the VEGF pathway mediated these mechanisms. mRNA and protein levels of AM increased in human osteosarcoma SOSP-F5M2 cells under hypoxic conditions [246]. AM blunted hypoxic-induced apoptosis via the CLR-RAMP receptor; this antiapoptotic effect was due to the upregulation of the Bcl-2 expression, which partially occurred via the activation of the MEK/ERK1/2 signaling pathway [246]. The increased Bcl-2 expression mediated by AM was inhibited with U0126, a selective inhibitor of MEK or AM_22–52_ (an AM-specific receptor antagonist) [246]. Thus, AM is a survival agent in osteosarcoma cells, and the targeting of this peptide could be a valuable antitumor strategy to promote apoptosis in these cells.

#### 3.2.21. Ovarian Cancer

##### Adrenomedullin

AM, CLR, and RAMP2/3 are present in ovarian cancer, and a high AM level has been related to the tumor stage [247,248]. *AM* gene expression has been studied in 60 cases of epithelial ovarian cancer. A correlation between this expression and the histological grade, lymph node metastasis, and prognosis has been reported. This correlation was not observed between *AM* gene expression and the disease stage, histological subtype, residual tumor mass after initial surgery, and patient’s age at diagnosis [249,252]. *AM* gene expression may define a more aggressive tumor phenotype. Moreover, AM was observed in carcinoma cells (cell membrane, cytoplasm) and endothelial cells of the tumor stroma [252]. Ovarian tumor cells synthesize AM, as well as express AM mRNA for both ligand and receptor, and the estrogen receptor alpha/beta ratio was higher in tumors than in other tissues; a correlation was observed between estrogen receptor alpha and estrogen receptor beta mRNA and AM mRNA expression in tumors [247]. It is known that AM metabolism (mRNA expression and secretion) is not under estradiol control and that AM and anti-AM antibodies show no growth actions in vitro. Nevertheless, the adjunction of anti-AM antibodies to estradiol exerted inhibitory effects on cells showing a high mitogenic activity [315]. AM mRNA expression was higher in granulosa tumor cells than in fibrothecomas and normal human ovaries. Activin A, prostaglandin E, follicle-stimulating hormone, and cAMP blocked *AM* gene expression in granulosa-luteal cells [316]. AM (mRNA and protein) was observed in ovarian epithelial carcinoma tissue, and the basic fibroblast growth factor, via the JNK-activator protein (AP)1 pathway, promoted AM expression in the CAOV(3) ovarian epithelial carcinoma cell line [317].

*AM* gene silencing blocked cell proliferation, increased the chemosensitivity in HO-8910 cells, decreased AM mRNA/protein expression, and downregulated p-ERK and Bcl-2 expressions [250]. AM is involved in the progression of epithelial ovarian cancer and cell migration by activating the integrin α5β1 signaling pathway [248]. AM also upregulated focal adhesion kinase (FAK)/paxillin phosphorylation. Cancer patients with a high AM expression showed a larger residual size of tumors after cytoreduction, a shorter disease-free and overall survival time, and a higher incidence of metastasis [248]. AM can polarize macrophages which favor, via activation of the RhoA signaling pathway, the migration of HO8910 ovarian cancer cells [318]. FAK and AM expressions were higher in epithelial ovarian cancer than in benign tumors or normal ovarian tissues. FAK can upregulate AM during the migration and invasion of epithelial ovarian cancer cells [249]. FAK and AM have been suggested as biomarkers to evaluate the prognosis and malignant potential in human ovarian cancer [249].

VEGF expression induced by AM was mediated through the JNK/AP1 pathway in HO-8910 epithelial ovarian carcinoma cells [319]. AM expression positively correlated with HIF-1α, VEGF, or microvessel density in epithelial ovarian cancer [251]. AM is an upstream molecule of VEGF/HIF-1α that favored angiogenesis after upregulating both factors in CAOV3 epithelial ovarian cancer cells [251].

#### 3.2.22. Pancreatic Cancer

##### Adrenomedullin

The expression of AM, CLR, and RAMP2/3 has been reported in pancreatic cancer, and a high level of AM is associated with a decrease in disease-free survival [253,254]. AM blocked glucose-stimulated insulin secretion from β-cells, and this effect was attenuated by AM knockdown [255]. AM overexpression led to glucose intolerance, and plasma AM level was higher in pancreatic cancer patients than in control patients or patients with diabetes [255]. Moreover, AM level was higher in pancreatic cancer patients with diabetes than in pancreatic cancer patients without diabetes [255]. The increased circulating AM in insulinoma is related to the neoplastic phenotype [256]. Pancreatic cancer promotes a paraneoplastic β-cell dysfunction by shedding AM(+)/CA19-9(+) exosomes into circulation, which block, via AM-induced endoplasmic reticulum stress, insulin release [320]. Moreover, pancreatic cancer exosomes induced lipolysis in subcutaneous adipose tissue [321].

AM and its receptor are expressed in pancreatic cancer cells, and AM is involved in the proliferation of these cells and metastasis [263]. AM increased the invasiveness of some pancreatic cancer cells, as well as regulated angiogenesis; AM is induced under hypoxic conditions, slightly upregulates VEGF release, and is overexpressed in pancreatic ductal adenocarcinoma, which could serve as a potential tumor marker [253]. AM mRNA/protein expressions increased in human pancreatic cancer samples compared to healthy individuals [255]. AM has been observed in the neoplastic epithelium of human pancreatic adenocarcinomas (43 of 48), and it was expressed in all eight pancreatic cancer cell lines studied; in addition, AM receptor silencing in pancreatic cancer cells inhibited AM-induced cell growth and invasion [257]. Thus, AM increases pancreatic cancer cell aggressiveness. AM antagonists decreased the growth of pancreatic cancer cells, and the diameter of blood vessels was smaller in tumors treated with AM antagonists than in those treated with AM N-terminal fragment (AM1-25) [254].

AM receptors mediate the AM stimulatory action on pancreatic cancer cells and endothelial and stellate cells. The silencing of AM receptors decreased both the growth of pancreatic ductal adenocarcinoma cells and the polygon formation of endothelial cells in vitro [263]. Pancreatic ductal adenocarcinoma is characterized by the stromal infiltration of myelomonocytic cells (which express all components of the AM receptor). This infiltration is associated with a poor prognosis [259]. Pancreatic cancer cells release AM, which, after activating the eNOS, PI3K/Akt, and MAPK signaling pathways and raising the expression of MMP-2, increased the migration/invasion of myelomonocytic cells [259]. AM also favored the migration of myelomonocytic cells by increasing ICAM-1 and VCAM-1 expression in endothelial cells and promoted the expression of protumor phenotypes in myeloid-derived suppressor cells and macrophages [259].

Desmoplasia is a prominent pathological characteristic of pancreatic cancer regulated by MYB [322]. MYB, a cellular progenitor of the v-Myb oncogene, encodes an oncogenic transcription factor that regulates gene expression (e.g., *AM* gene) [322]. AM controlled the growth of pancreatic tumor cells and pancreatic stellate cells in this disease [322]. Lastly, when PAN02 pancreatic cancer cells were injected into the spleens of mice, spontaneous liver metastasis occurred [258]. In this model, selective activation of RAMP2 and inhibition of RAMP3 could suppress tumor metastasis [258].

##### Adrenomedullin 2

AM2 is a tumor angiogenic factor in pancreatic cancer, regulating MAPK, ERK, Akt, PI3K, and VEGF pathways. The total cellular AM2 level is a poorer predictor of survival in this disease (patients with pancreatic cancer and a high expression of AM2 showed a median survival of 10 months shorter than the remainder of the cohort studied). Hence, this level could be a biomarker for predicting survival [260].

##### Amylin

AMY level is an early feature of pancreatic cancer [323] and has been suggested as a marker for neuroendocrine tumor growth [324]. However, later studies reported that a high plasma AMY level is not a good marker for detecting pancreatic cancer [325,326,327]. AMY, a glycolysis inhibitor, prevents BRAF and RAS oncogene-induced senescence in human cells. This connection between oncogene-induced senescence and AMY is an important link between metabolism and cancer since metabolic reprogramming is a cancer hallmark [328].

A relationship between insulin and AMY has been suggested regarding human AMY toxicity in pancreatic β-cells [329]. AMY overexpression seems to cause amyloid deposition in the islets of type 2 (non-insulin-dependent) diabetic patients and insulinomas [330]. In type 2 diabetes, the deposit of β-cell-toxic islet amyloid is a characteristic feature. The main amyloid component is AMY, but heparan sulfate proteoglycan is also present [331]. Heparan sulfate binds to AMY, promoting AMY conformational changes and accelerating the formation of fibrils [331]. This interaction could be a therapeutic target to increase β cell survival. Type 3 diabetes mellitus is a harbinger of pancreatic cancer in at least 30% of patients [332].

Insulinomas express AMY [264], and AMY or islet amyloid polypeptide (IAPP) has been detected in endocrine tumors (e.g., pancreatic tumors producing insulin or glucagon) and human pancreas [333]. This result means that AMY could be related to amyloid deposition in endocrine tumors. Silibinin inhibited the cytotoxic effects mediated by AMY on the INS-1 insulinoma cell line [334]. Glucose increases AMY expression in normal pancreatic β-cells [329], and an inhibitor of β-cell glucokinase mannoheptulose and glucose analog 2-DG inhibited the effect of glucose on AMY mRNA; this means that glycolysis is necessary for AMY mRNA accumulation [329]. Long-standing diabetes increases the risk of pancreatic cancer, and recent-onset diabetes is also associated with a risk increase [332]. The plasma AMY level is augmented in nondiabetic patients with pancreatic cancer, but this level is low in patients with diabetes [261]. AMY promotes apoptosis in human pancreatic islet β-cells via the stimulation of c-Jun expression/activation [335]. It seems that dimerization and co-expression of c-Jun and c-fos or ATF-2 are essential to induce apoptotic mechanisms.

##### Calcitonin Gene-Related Peptide

CALCA (αCGRP) and CALCB (βCGRP) methylation increased in pancreatic adenocarcinoma cells, and, under that condition, p-CREB and p-AKT levels decreased [262].

#### 3.2.23. Pituitary Adenoma

##### Adrenomedullin

AM expression was lower in anterior pituitary tumors than in normal glands [265].

##### Amylin

A study reported that AMY is not a suitable marker for pituitary adenomas; however, this must be confirmed in future studies [325].

#### 3.2.24. Prostate Cancer

NKX3.1 is a prostate-specific transcription factor related to prostate development and tumor progression. NKX3.1 deletion leads to premalignant prostate lesions [180]. *RAMP1* is an NKX3.1 target gene, upregulated in prostate cancer, and RAMP1 knockdown in prostate cancer cell lines showed a decrease in colony formation, numbers of cells in the S phase, and cell proliferation [180]. Moreover, ERK1/2 phosphorylated level was reduced, linking RAMP1 to the MAPK pathway [180].

##### Adrenomedullin

CLR and RAMP2/3 have been reported in prostate cancer, and a high level of AM has been associated with a high Gleason score [266,267]. Plasma AM2 level has been associated with prognostic factors (tumor node metastasis, 5 year metastasis) in prostate cancer patients [180]. AM is widely expressed in prostate carcinomas and normal prostates, and the peptide was observed in neurons of the prostate ganglia and endothelial, stroma, and secretory cells [268]. Prostate cancer cells are derived from prostatic epithelial cells in which the expression of AM has been observed. AM, through autocrine and paracrine mechanisms, suppressed prostate cancer cell malignancy [336]. AM (overexpressed or exogenously added) inhibited prostate cancer cell growth. This blockade was due to ERK1/2 activation and not cAMP intracellular level changes; this study was performed in PC-3, DU 145, and LNCaP prostate cancer cells expressing the CLR/RAMP-2 receptor complex [269]. This overexpression resulted in the dysregulation of approximately 100 genes involved in apoptosis, cell cycle, extracellular matrix, cell adhesion, and cytoskeleton [337]. For example, AM upregulated genes (e.g., RUNX-3, IGF-BP6, and GADD45) are involved in cell growth arrest [337]. However, other studies showed that AM, mediated by the AM2 receptor (CLR/RAMP3) subtype, promoted the growth of the human prostate, being the peptide involved in developing epithelial cell malignant phenotype. After serum deprivation, AM blocked apoptosis in DU 145 and PC-3 cells, but not in LNCaP cells [270]. Moreover, an AM-induced mechanism of tumor sensitization has been reported, upregulating the interleukin-13R alpha 2 chain, a target for the specific recombinant chimeric cytotoxin interleukin-13-PE38 [337]. Interleukin-13 receptor alpha 2 (IL-13Ralpha2) is a tumor antigen, and a high-affinity interleukin-13-binding subunit is known to be amplified in human tumor cell lines and tumors [338]. AM upregulated IL-13Ralpha2 in a human prostate tumor cell line, indicating that the peptide and interleukin-13 are closely related to each other. AM is involved in sensitizing prostate tumors to the interleukin-13R-directed therapeutic agent.

AM was located in the carcinomatous epithelial compartment, and androgen-independent PC-3 and DU145 cell lines derived from human prostate cancer synthesized and released AM, which acted as a growth factor on DU145 cells [266]. Advanced and metastatic prostate carcinomas treated with antiandrogenic therapy become, in general, refractory to the treatment. It has been reported that pre-proAM and peptidyl glycine alpha-amidating monooxygenase expressions are androgen-independent in the human prostate [339]. The previous enzyme is involved in AM amidation, which must be fully active. No relation between anti-androgenic treatment or Gleason score and AM expression has been reported [339]. AM, upon androgen ablation, is involved in hormone-independent tumor growth and lymphangiogenesis and neoangiogenesis [267]. Neuroendocrine differentiation is an early marker associated with developing androgen independence in prostate cancer. It has been reported that AM is a mediator in such differentiation and that its production is vital for tumor resurgence following androgen ablation [340].

AM is involved in the spread of prostate tumors to the bone and can also influence localized levels of RANKL in the bone to favor tumorigenesis [180]. AM, via transient receptor potential channel (TRPV2) translocation to the plasma membrane, promoted prostate and urothelial cancer cell migration and invasion, increasing resting Ca^++^ level [271]. Moreover, AM induced the phosphorylation of FAK and β1 integrin in prostate cancer cell lines, which raised these cells’ invasive and migratory capacity [180]. Smooth muscle cells migrated when these cells were cultured with conditioned media from lymphatic endothelial cells; this effect and the formation of lymphatic vasculature were inhibited when the action of AM was blocked [267]. This observation means that AM promotes tumor lymphangiogenesis [180].

##### Adrenomedullin 2

AM2, upregulated under hypoxic conditions, is involved in prostate cancer progression by stimulating cancer cell migration and angiogenesis [305,341]. Plasma AM2 levels were higher in patients with prostate cancer than in healthy individuals, and those patients showing a Gleason score ≥7, organ unconfined, seminal vesicle invasion, tumor node metastasis stage T2, positive lymph node, or extra-prostatic extension showed a higher level of AM2 [272]. The elevated plasma AM2 level is independently associated with long-term recurrence and distant metastasis in prostate cancer [272]. AM2 has been suggested as a prognostic, predictive biomarker for 5 year metastasis and 5 year progression.

##### Calcitonin Gene-Related Peptide

CGRP is expressed in the prostate’s neuroendocrine cells and sensory nerves [180]. Neuroendocrine cells frequently differentiate in prostate cancer, the peptide increases prostate cancer cells’ invasive and migratory capacity, CGRP serum level correlates with prostate cancer progression in humans, and CGRP receptor mediates prostate cancer cell metastasis to the bone [274,342,343]. Neuronal CGRP promotes prostate tumor growth in the bone microenvironment and the femur of experimental animals; this effect was blocked with CGRP antagonists [275]. Moreover, CGRP activated the signal transducer and activator of transcription 3 (STAT3) and ERK signaling pathways in prostate cancer cells [275]. Serum CGRP level was higher in patients with higher histological grade and clinical stages than in patients with lower grade and stages [273,274]. Thus, in untreated prostate cancer patients, serum CGRP level could be a predictor of grading and staging; this does not occur in patients with prostate cancer who received hormonal treatment since this treatment influences the level of CGRP [274].

#### 3.2.25. Renal Carcinoma

##### Adrenomedullin

AM, CLR, RAMP2, and RAMP3 have been reported in renal cancer, and a high level of CLR has been associated with a high tumor grade [276,277,313]. Both VEGF and AM were observed in the cytosol of human renal carcinoma cells, while VEGF and AM mRNAs were also detected in this carcinoma [278]. This outcome suggests that AM facilitates tumor angiogenesis in conjunction with VEGF and favors renal carcinoma cell growth. Human clear cell renal cell carcinoma displays mutations in the tumor suppressor von Hippel–Lindau protein, leading to a high expression of HIF-1 [276]. Higher AM tissue levels have been reported in human clear cell renal cell carcinomas than in other kidney tumors, but plasma AM levels cannot be used as a tumor marker for this disease [276]. AM, CLR, and RAMP2 were observed in the carcinomatous epithelial compartment of chromophobe renal cell carcinoma (CRCC), and RAMP3 was only found in the inflammatory cells that infiltrated tumors [313]. AM mRNA level was higher in CRCC than in normal renal tissue, and AM mRNA expression correlated with VEGF-A mRNA expression in CRCC [313]. CRCC BIZ and 786-O cells expressed and released AM, the peptide promoted cell proliferation, migration, and invasion, and these actions were mediated by CLR/RAMP2 and CLR/RAMP3 [313]. Moreover, a high AM mRNA level has been related to an increased risk of relapse after curative nephrectomy for CRCC.

#### 3.2.26. Thymic Lymphomas

##### Amylin

Pramlintide, a synthetic analog of AMY, promoted tumor regression in p53-deficient thymic lymphomas [279]. AMY, via CT receptor and RAMP3, blocks glycolysis, promotes the formation of reactive oxygen species, and induces apoptosis [279]. Thus, an antitumor strategy targeting p53-deficient cancers could be developed.

#### 3.2.27. Thyroid Cancer

##### Adrenomedullin 2

AM2 is secreted from tumor cells under mitochondrial stress resulting from overnutrition, and it is involved in developing thyroid cancer [280]. AM2 upregulation was increased after the activation of the mitochondrial stress response pathway in tumor cells, and AM2 expression was increased in obese patients with thyroid cancer showing locoregional recurrence, a high prevalence of lymph node metastasis, and larger tumor size [280]. High levels of circulating AM2 and tumor cell AM2 expression have been associated with aggressive pathological parameters (e.g., body mass index) in patients with thyroid cancer. AM2 plays an essential role in the nexus of obesity and thyroid cancer, and the level of AM2 could be used as a biomarker for predicting thyroid tumor progression [280].

##### Amylin

Medullary thyroid carcinoma patients (seven of 10) showed high levels of AMY, and the peptide was observed by immunohistochemistry in four of 12 medullary thyroid carcinomas and two of five lymph-node metastases [281]. Serum insulin and plasma AMY levels were correlated in both controls and patients with medullary thyroid carcinoma. However, the slope of the regression line was significantly higher for medullary thyroid carcinoma patients [281].

##### Calcitonin Gene-Related Peptide

*CGRP* gene expression has been reported in thyroid carcinoma; CGRP is released from medullary thyroid carcinoma cells, and plasma CGRP level has been suggested as a marker for medullary thyroid carcinoma [237,282,283].

#### 3.2.28. Uterine Cervical Carcinoma

##### Adrenomedullin

High AM and AM mRNA expressions were observed in the cytoplasm of invasive cervical squamous carcinoma cells. Both expressions were more prominent in the stromal cells adjacent to early invasive carcinoma cells than in the carcinoma cells [284]. Consequently, the peptide appears involved in the growth and invasion of uterine cervix squamous carcinoma cells. Another study reported that, in invasive uterine cervix squamous carcinoma, AM and Bcl-2 were involved in promoting malignant progression and in selecting carcinoma cells resistant to apoptosis [285].

#### 3.2.29. Vascular Tumors

##### Adrenomedullin

The proliferation of endothelial cells characterizes vascular tumors (e.g., capillary haemangioma and Kaposi’s sarcoma), and these tumors express more CLR than that expressed in the adjacent normal endothelium [344]. Therefore, the proliferation of endothelial cells and the neoplastic vascular growth seem to be mediated by AM.

Figure 14, Figure 15 and Figure 16 summarize the main findings regarding the involvement of AM, AM2, AMY, and CGRP in cancer.

## 4. Antitumor Therapeutic Strategies Based on the Modulation of Peptidergic Systems

### 4.1. Tachykinin Peptide Family

No NK-2R or NK-3R antagonist has been approved for clinical practice [14], although several NK-2R or NK-3R antagonists have been tested in clinical trials [16]. Unfortunately, although NK-2R/NK-3R antagonists were safe, these trials were abandoned due to the lack of efficacy. The ineffective results could also have been due to a non-appropriate selection of patients and endpoints in clinical trials and a lack of knowledge regarding the molecular interaction between tachykinin ligands and NK receptors [17]. Moreover, the ineffective effects could also be due to a non-appropriate selection of patients and endpoints in clinical trials and to a lack of knowledge regarding the molecular interaction between tachykinin ligands and NK receptors [17]. NK-2R antagonists (e.g., nepadutant (MEN-11,420) blocked tumor cell proliferation promoted by NKA in breast carcinoma [139]. Metastatic and non-metastatic breast cancer cell proliferation was blocked by inhibiting the action of SP, at NK-2R, with NK-2R antagonists (GR-159,897); however, this action was less prominent than that observed with NK-1R antagonists (RP-67,580) [140]. This finding suggests that breast cancer cells respond differently to NK-2R and NK-1R antagonists. Moreover, these cells reacted differently to NK-2R and NK-1R agonists (e.g., SP) [140]. NK-2R antagonists enhanced the secretion of macrophage inflammatory protein-2 and increased p38 phosphorylation [140]. This study also reported that SP altered neither breast cancer cell proliferation nor the effects mediated by NK-2R and NK-1R antagonists on these cells [140]. The latter does not agree with the results reported in most of the performed studies. This discrepancy must be investigated. SB-222,200, a selective NK-3R antagonist, inhibited tumorigenesis and osteolytic lesions promoted by the oral squamous tumor cells that invade the bone matrix, inducing bone destruction [156]. GR-138,676 shows little activity at NK-2R. It can be used as an NK-3R antagonist, despite its activity at NK-1R [345]. NK-3R has been suggested as a potential antiangiogenic target because NKB exerted anti-angiogenesis effects. NKB analogs have shown antimigration and antitumor activities in vivo with no significant side-effects [346].

### 4.2. Calcitonin/Calcitonin Gene-Related Peptide Family

#### 4.2.1. Adrenomedullin

AM ligand/receptor overexpression occurs in colonic cancers, and antibodies directed against both targets decreased tumor growth [204,205,206,347,348,349,350]. The peptide fragment AM_22–52_ [254], polyclonal antibodies against AM [217,308] or its receptor [350], monoclonal antibodies [351], and small interfering RNAs [257] regulate AM expression and actions [208]. Small molecules (e.g., 16311, a negative AM modulator; 145425, a positive AM modulator) also regulate the effects mediated by AM. For example, 145425 reduced tumor burden, colon weight/length ratio, and tumor growth in mice [208]. AM promotes cell proliferation and survival, alters the cell phenotype more aggressively, and increases vascularization [352]. Tumors are, in general, the main source of excessive AM production since AM levels returned to normal levels when the tumor was removed after surgery [353]. The blockade of AM receptors or lowering AM amounts are antitumor strategies to decrease the tumor mass [175]. Many preclinical studies have shown reduced tumor cell proliferation, metastasis, and angiogenesis after applying AM receptor antagonists, AM receptor interference, or AM-neutralizing antibodies [352]. Thus, a cocktail of antibodies directed against CLR and RAMP2/3 exerted an antitumor effect against glioblastoma, mesothelioma, and lung and colon cancer, and peptide antagonists such as AM_22–52_ also showed an antitumor action against melanoma, mesothelioma, and renal, ovarian, breast and pancreatic cancer [217,350,354]. Moreover, small-molecule antagonists targeting the AM ligand (NSC 16311) or AM2 receptors (2, 2-dimethyl-N-((2-(methylaminomethyl) phenyl) methyl)-N-(2-oxo-2-(((2 R)-2′-oxospirol (1, 3-dihydroindene-2, 3′-1 H-pyrrolo (2, 3-b) pyredine)-5-yl) amino) ethyl) propanamide) respectively exerted antitumor effects against breast cancer and breast/pancreatic cancer [236,355,356]. Cancer patients show, in general, a high level of circulating AM, and this high level has been associated with the worst prognosis [353]. AM-1 and AM-2 receptors are involved in cancer development; the former also regulates blood pressure. Thus, using AM-1 receptor antagonists as antitumor agents is not possible since these antagonists increase blood pressure [357]. However, AM-2 receptor antagonists are promising agents as antitumor drugs because they show good pharmacokinetic properties, no serious side-effects, and 1000-fold selectivity over the AM-1 receptor [355,357]. In this sense, it is crucial to know which receptors are involved in the effects mediated by the CT/CGRP peptide family in cancer. This knowledge will help to develop selective receptor antagonists [175]. Potent small-molecule AM-2 receptor antagonists have been synthesized but retain activity against the CGRP receptor [355]. These compounds blocked tumor growth and extended life in an experimental animal model of pancreatic cancer [355]. This finding is important and must be developed as an anticancer strategy, in addition to increasing the knowledge of AM receptor pharmacology. The importance of investigating the residue differences between the AM receptors has also been highlighted to develop selective AM-2 receptor antagonists [357].

AM antagonist peptide (AM_22–52_) decreased the growth of pancreatic cancer cells in vivo. It has been reported that intramuscular or intratumoral transfer of naked DNA encoding AM_22–52_ might be a promising strategy for treating human cancers [358]. This strategy eliminates CD31-positive cells from the tumor, which means that neo-vascularization is completely inhibited [358]. A constructed AM antagonist expression vector decreased tumor growth and microvessel density in renal cell carcinoma [359]. Moreover, AM inhibition blocked sunitinib-resistant renal cell carcinoma growth by targeting the ERK/MAPK pathway; this means that combination therapy with AM receptor antagonists and sunitinib is a promising strategy against renal cell carcinoma [360].

Tumor development is highly dependent on the formation of both lymphatic (lymphangiogenesis) and blood (angiogenesis) vessels from pre-existing ones; these mechanisms are regulated by AM and VEGF, which are released from cancer and tumor stroma cells (e.g., endothelial cells, pericytes, fibroblasts, and macrophages) [352]. Thus, the inhibition of both angiogenesis and lymphangiogenesis mechanisms is a sound antitumor strategy. The AM/RAMP2 system controls vascular integrity. It has been reported that the deletion of RAMP2 favored vascular permeability and the formation of pre-metastatic niches in distant organs by altering the vascular structure and promoting inflammation [361]. As a result, the AM/RAMP2 system regulates vascular integrity, and this system could be a promising antitumor therapeutic target to block metastasis. AM plays an essential role as a crosstalk agent integrating mast and tumor cell communication [362]. AM favors the release of beta-hexosaminidase or histamine from human mast cells, which are also involved in angiogenesis; this was inhibited with anti-AM monoclonal antibodies [362]. Alpha-AMR (an antibody against the AM receptor) decreased the growth of HT-29 colorectal cancer cells and U87 glioblastoma cells in vitro, as well as the growth of colon, lung, and glioblastoma tumors in vivo [338]. Alpha-AMR increased apoptosis in cancer cells, depleted endothelial and pericyte cells, disrupted tumor vascularity, and decreased microvessel density [338]. Thus, alpha-AMR blocked angiogenesis and tumor growth. Moreover, the use of a polyclonal antibody designed to block all the receptor components of AM (RAMP2, RAMP3, and CLR) decreased both proliferation and invasion of prostate cancer cells, tumor weight, metastasis, and lymphatic vasculature [180]. Notably, the reduction in lymphatic vasculature was observed in the tumor, not in the tissue surrounding the tumor [180].

*AM* gene silencing decreased the expression of both AM mRNA and protein, blocked cell proliferation, and increased the chemosensitivity of HO8910 ovarian cancer cells via downregulation of Bcl-2/p-ERK expressions [250]. Anti-AM treatment and Bcl-2/ERK inhibited expressions are promising strategies to treat ovarian cancer [250]. Carriers of a single-nucleotide polymorphism (rs4910118) in the proximity of the *AM* gene showed lower levels of circulating AM; patients with cancer (breast, lung) showed a lower frequency than healthy donors, and carriers of the minor allele showed a lower risk of developing cancer than those homozygotes for the major allele [363]. The authors concluded that carriers of rs4910118 close to the *AM* gene are protected against cancer.

The AM/RAMP2 system is involved in tumor angiogenesis; liver metastasis (PAN02 pancreatic cancer cells were administered into the spleen) increased in vascular endothelial cell-specific RAMP2 knockout mice, and liver metastasis was suppressed in RAMP3^−/−^ animals in which the number of podoplanin-positive cancer-associated fibroblasts decreased in the periphery of tumors at metastatic sites [258]. RAMP3 deficiency cancer-associated fibroblasts inhibited cell proliferation/migration and metastasis, and the activation of RAMP2 in RAMP3^−/−^ mice blocked tumor growth and metastasis [258]. Moreover, podoplanin upregulation in RAMP2^−/−^ animals augmented malignancy, and podoplanin downregulation in RAMP3^−/−^ mice decreased malignancy [258]. The observation means that RAMP2 activation and RAMP3 inhibition can suppress metastasis, and that deficiency of the AM/RAMP3 system inhibited metastasis via the modification of cancer-associated fibroblasts. Acylated truncated AM/AM2 analogs of 27–31 residues showed a potent antagonistic action toward CLR/RAMP1, and non-acylated analogs showed minimal activity [364]. The authors of the study reported that a lysine residue in a 17-amino-acid analog, named antagonist 2–4, is important for increasing its antagonistic activity; they also showed that the analog, showing a 12-amino-acid binding domain of CGRP and AM sequence motif, exerted a potent CLR/RAMP1-inhibitory effect, and that a chimeric analog, constituted by an AM antagonist and a somatostatin analog, exerted a dual effect on CLR/RAMP and somatostatin receptors [364]. The blockage of CLR/RAMP inhibited tumor growth and metastasis, and it can be concluded that somatostatin-AM antagonist analogs could be applied to treat tumors [364].

In sum, current AM-based antitumor therapeutic strategies have been focused on targeting AM with blocking antibodies (e.g., adrecizumab, mAb-C1, and MoAb-G6) or AM receptor antagonists (e.g., NSC-16311 and NSC-37133, both small molecules with antitumor properties). They show more favorable pharmacokinetic characteristics than antitumor peptides such as AM_22–52_; this is important since using specific anti-RAMP2 or anti-RAMP3 antibodies is crucial to selectively inhibit AM-1 or AM-2 receptors [352]. Moreover, it has been reported that a ribozyme diminished the expression of AM, although this strategy shows significant disadvantages from a pharmacokinetic point of view [352].

#### 4.2.2. Adrenomedullin 2

AM2 is involved in vascular remodeling processes and angiogenesis; hence, the peptide is an important target for developing angiogenesis-based antitumor strategies [305]. AM2 increases tumor blood perfusion and promotes quiescent endothelial cells to proliferate by restraining endothelial cell response to VEGF; hence, the excessive vessel sprouting is blocked, and the vascular lumen is increased [365]. AM2 also favors the formation of a signaling complex containing CLR/β-arrestin1/Src in endothelial cells and promotes its internalization into the cytoplasm via a clathrin-dependent manner to activate downstream ERK1/2 pathway; this action was not inhibited by endothelial cell contact [365].

AM2 blockade with neutralizing antibodies/antagonist peptides inhibited the growth of tumor cells by promoting apoptosis, Bcl2/Gli1 downregulation, and caspase-8 cleavage [231]. Significantly, administering anti-AM2 monoclonal antibodies not only inhibited tumor growth but also increased the antitumor activity of temozolomide [222]. By using anti-AM2 antibodies, ERK1/2 phosphorylation was inhibited in endothelial cells and, in the same cells, a decreased expression of vascular endothelial cadherin and dissociation of vascular endothelial cadherin/vascular endothelial growth factor receptor 2/phosphoinositide 3-kinase complex occurred, leading to the internalization and phosphorylation of vascular endothelial growth factor receptor 2 and the blockade of the PI3K/Akt pathway [305]. Moreover, anti-AM2 monoclonal antibodies blocked glioblastoma multiforme growth and significantly enhanced the activity of temozolomide [222].

#### 4.2.3. Amylin

The p53 family promotes tumor suppression, and deletion of the ΔN isoforms of p63 or p73 led to metabolic reprogramming and regression of p53-deficient tumors via upregulation of the *AMY* gene [279]. AMY is involved in tumor regression and, via the CT receptor/receptor activity modifying protein 3, promotes apoptosis, induces reactive oxygen species, and blocks glycolysis [279]. Pramlintide, a synthetic analog of AMY, stimulated tumor regression in p53-deficient thymic lymphomas, representing a novel strategy to target p53-deficient cancers [279]. Pramlintide also exerted an antiproliferative action against colorectal cancer cells, and its coadministration with classic chemotherapeutics increased cytotoxicity [366].

#### 4.2.4. Calcitonin Gene-Related Peptide

CGRP signaling, through the CT receptor, increased chemotherapy resistance and stem cell properties in acute myeloid leukemia, and olcegepant, a CGRP antagonist, decreased key stem cell properties and leukemic burden [182]. The CGRP/CT receptor system could be an antitumor target against acute myeloid leukemia. Moreover, the CGRP8-37 peptide antagonist exerted an antitumor action against prostate cancer [367].

CGRP augmented cytotoxic CD8^+^ T cell exhaustion, limiting their capacity to eliminate melanoma cells [368]. CGRP antagonism of the receptor RAMP1 or pharmacological silencing of nociceptors decreased tumor-infiltrating leukocyte exhaustion and B16F10 melanoma cell growth [368]. Thus, reducing CGRP release from tumor-innervating nociceptors is a valuable strategy to improve antitumor immunity via eliminating the immunomodulatory actions exerted by CGRP on these cytotoxic T cells [368].

## 5. Future Research

Previous sections showed how both peptide families reviewed here are involved in cancer progression and how several promising antitumor strategies could be developed. This knowledge is crucial, highlighting that the same peptide favors the development of many different tumors and that a common antitumor approach could be possible. Tumor size, higher tumor aggressiveness, malignancy, increased relapse risk, worse sensitivity to chemotherapy agents, and poor prognosis are associated with the peptidergic systems reviewed here. Peptides are widely involved in cancer development (mitogenesis, migration, invasion, metastasis, angiogenesis, lymphangiogenesis, and antiapoptotic effect). Some peptides exert an antiproliferative action against cancer cells, while others promote proliferative and antiproliferative effects. Consequently, peptidergic systems are promising antitumor targets to open up new cancer research lines, select new molecular targets (e.g., receptors), improve cancer diagnosis (peptidergic systems as prognostic biomarkers), and explore new antitumor strategies using new compounds (e.g., peptide antagonists) that could specifically destroy cancer cells and block angiogenesis alone or in combination with radiotherapy/chemotherapy. In the SP/NK-1R system, it has been demonstrated that the combination therapy of NK-1R antagonists with chemotherapy/radiotherapy increased the antitumor activity by chemosensitization and radiosensitization, as well as decreased the serious side-effects mediated by chemotherapy/radiotherapy (e.g., neurogenic inflammation, nephrotoxicity, neurotoxicity, hepatotoxicity, and cardiotoxicity) [4].

The involvement of NKB/NK-3R in cancer has been much less studied than NKA/NK-2R. Thus, information is lacking regarding the NKB/NK-3R system (e.g., NKB/NK-3R expression and mitogenesis, migration, and metastasis mediated by NKB in many tumors, as well as involvement of NK-3R in the viability of these cells). This critical research line must be developed in the following years to know which type of tachykinin receptor (NK-1R, NK-2R, and NK-3R) is expressed in the same tumor cell, and how SP, NKA, and NKB interact and regulate tumor cell mitogenesis and migration. This knowledge is also crucial to develop effective antitumor treatments. In most tumors (e.g., astrocytoma, insulinoma, neuroblastoma, phaeochromocytoma, schwannoma-derived cells, lung cancer, and leiomyoma), knowledge on the involvement of NKA/NKB is lacking, scarce, or incomplete. For example, NK-3R expression has been reported in astrocytoma cells [145]; however, it is not known whether NKB exerts a proliferative action in these cells.

Regarding insulinoma, it is only known that cancer cells release NKA and express the *pre-protachykinin A* gene [146,147]; in the pathological events observed in women suffering from leiomyomata, the involvement of NKB/NK-3R must be established [91]. However, in a few cancers (e.g., breast cancer and colorectal cancer), the involvement of NKA has been better studied. For example, in breast cancer, NKA promotes the migration of cells and exerts a proliferative action on tumor cells that are blocked with NK-2R antagonists [92,139,140,141]. In colorectal cancer, an important point must be investigated [142,143], i.e., why NKA did not exert a growth-regulatory action on the colon cancer cell line HT 29 [168]. The answer may be that the tachykinin receptor type and the activated signal transduction pathways depend on the coupled G protein type.

Moreover, NKB/NKB analogs exerted anti-angiogenesis and antimigration/antitumor actions [346]. It must be investigated whether these actions mediated by NKB are observed in different tumors since NKA favored through NK-1R the proliferation of glioma cells [144]. This is a crucial point because NKA exerted its physiological actions not exclusively via NK-2R but also via NK-1R. Moreover, it is essential to remark that NKA, NKA_3-10_, and NKA_4-10_ have been reported in ileal metastatic carcinoid tumors; these tumors can release different amounts of several tachykinins contributing to individual differences [165]. Knowing the physiological actions mediated by the unmodified peptide and its fragments is an important point that must be studied in other tumors.

Studies have reported that NKA/NKB blocked the growth of small-cell lung cancer cells [20]. This is a significant point that must also be investigated in depth to develop new antitumor strategies since NKA exerts a proliferative action on other tumor cells, and it is widely known that SP, another member of the tachykinin peptide family, promotes the proliferation of many different human cancer cell lines [5]. Oral squamous tumor cells do not express NKB; however, NKB appears in sensory nerves in the mandible. The finding suggests the release of the peptide from these nerves and that NKB could regulate the proliferation of tumor cells. The above must be confirmed to establish a clear relationship between tumor cells and the nervous system. A crucial point is the use of peptidergic systems as predictive biomarkers. In this line, elevated plasma NKA concentration has been suggested as a biomarker of prognosis in patients suffering from midgut carcinoid tumors [150] and to predict survival in patients with small bowel neuroendocrine tumors [161,163]. The result must be confirmed in future studies. Cancer biomarkers are crucial since some tumors are challenging to diagnose and because monitoring peptide plasma levels could be handy in selecting patients showing a poor prognosis and requiring urgent therapeutic intervention.

Compared with NKA/NKB, many more studies have been focused on the involvement of the CT/CGRP peptide family in cancer. It is established that members of this family favor tumor cell proliferation, metastasis, angiogenesis, and lymphangiogenesis, prevent apoptosis, and suppress the immune system [175,211,369,370]. AM level has been correlated with cancer severity [6,104,175]. AM is the most studied member of the CT/CGRP peptide family, followed by AM2, whereas AMY and CGRP are less studied. Thus, much effort must be performed to increase the knowledge of this family in cancer and, in particular, to decipher the roles played by AMY and CGRP. An important point that must be developed and clarified is the involvement of AM in the post-chemotherapy persistence of solid tumor cells; in leukemia, whether targeting CLR prevents relapse or not, and the methylation patterns of certain genes in this disease must be investigated [183,186]. Additional studies must be performed to confirm that the blockade of the CT receptor/CGRP system is a useful antitumor strategy to treat leukemia [136]. Two important findings have been reported in phaeochromocytoma: AM inhibited the proliferation of tumor cells, and circulating AM was increased in patients suffering from the disease [190]. Is this increase an endogenous antitumor strategy directed against the tumor? Moreover, in phaeochromocytoma, the involvement of AM2 in tumor growth must be elucidated, and the role of STAT-3 in invasion and metastasis must be investigated in tumors other than astroglioma [192,220]. The knowledge on the involvement of AM in bladder cancer is scarce [193]. Hence, additional experiments must be developed to confirm its involvement and fully demonstrate that AM is a potential antitumor target in this disease. AM is a tumor survival factor in breast cancer; therefore, the blockade of AM could be a promising antitumor strategy [99]. This must be confirmed, in addition to whether the level of circulating AM can be used or not as a predictor for lymph node metastasis in this disease. By contrast, AM_22–52_ decreased tumor cell proliferation, induced apoptosis, and disrupted tumor vasculature in breast carcinoma [197]. These findings are important since they highlight that tumor development was mediated by the unmodified peptide, whereas the contrary effect, i.e., an antitumor action, was favored by its fragment AM_22–52_ in breast cancer. Thus, the role played in cancer by peptide fragments belonging to the two peptide families studied here must also be elucidated. Moreover, AM, through its action on cancer cell EMT, decreased tumor cell invasion [196]; this finding means that AM could act as an antimetastatic agent in triple-negative breast cancer; however, the effect must be investigated and confirmed. AM mRNA expression has been described in choriocarcinoma [202]; however, it is unknown whether AM or other peptides belonging to the CT/CGRP peptide family exert a proliferative action on choriocarcinoma cells. Like choriocarcinoma, much information is lacking regarding the involvement of AM, AM2, AMY, and CGRP in certain tumors such as astroglioma, cutaneous nerve neuromas, neuroendocrine tumors, oropharyngeal squamous cell carcinoma, thymic lymphomas, Ewing sarcoma, uterine cervical carcinoma, and vascular tumors. It is important to remark that peptides belonging to the CT/CGRP peptide family favor the growth of different tumors; however, it is currently unknown whether CT/CGRP receptor antagonists exert an antitumor effect against these tumors. It is probably the case, but it is a crucial piece of information currently missing that deserves investigation. AM expression has been suggested as a predictive factor for tumor progression in patients with cancer; this is an important point that must be confirmed in the future. For example, AM expression has been suggested as a marker for predicting cancer-related death and a high risk of relapse in colorectal cancer patients with a curative resection [203]. It must also be confirmed whether plasma AM level can be used as a biomarker for predicting the long-term prognosis of patients suffering from nasopharyngeal carcinoma, whether AM expression can be used to evaluate the prognosis and malignant potential in human ovarian cancer, and whether AM2 expression can be used to predict survival in pancreatic cancer [240,249,260]. Additional experiments must be performed to confirm that plasma AM2 level is a prognostic biomarker for breast cancer patients [198]. Another important point is to know how the tumor microenvironment is involved in tumor progression. In this sense, tumor-associated macrophages, through AM, favor melanoma growth and angiogenesis, and the number of tumor-associated macrophages expressing and releasing AM in the tumor microenvironment has been associated with poor prognosis in this disease [239]. However, this knowledge is lacking in many other tumors and should be investigated in depth. It is also important to remark that AM inhibits prostate cancer cell growth [269], but it has been demonstrated that AM, through the AM-2 receptor, promotes the growth of the human prostate. This contradictory finding highlights that the physiological actions mediated by peptides could depend on the receptor type expressed by tumor cells. For this reason, it is crucial to know which receptors are expressed in cancer cells to develop specific antitumor strategies. After targeting AM, an important antitumor strategy has been demonstrated in glioma: miR-1297 blocked its expression and sensitized glioma cells to temozolomide treatment [221]. The aforementioned is an important finding and research field that must be developed in other tumors. AMY induces apoptosis in thymic lymphoma via the RAMP3/CT receptor [279]. Moreover, pramlintide, a synthetic analog of AMY, exerted an antiproliferative action against tumor cells, and its coadministration with chemotherapeutics increased cytotoxicity [366]. This knowledge is important for developing new antitumor strategies.

Extensive experimental evidence implicating peptidergic systems in cancer appearance, development, or treatment failure derives from studies in cancer cell lines isolated and maintained in culture. Furthermore, the literature provides numerous reports establishing the abnormal presence or absence of peptides and their receptors or mRNAs in different types of cancer cells extracted from tumors. Without a doubt, this information is relevant. However, new experimental approaches centered on analyzing dynamic and complex systems would provide relevant data concerning mechanisms triggered by peptidergic systems and their crosstalk signaling pathways under stress, eventually provoking cellular changes leading to uncontrolled growth and survival. Furthermore, determination of the influence of the peptidergic systems on the cell microenvironment would be ideal to unravel unknown factors that may explain the appearance and maintenance of cancer cells.

## 6. Conclusions

Due to the peptidergic systems’ crucial roles in cancer, a promising line of research is to study the involvement of peptides and their receptors in tumor progression to develop specific antitumor therapeutic strategies. Unfortunately, pharmaceutical companies generally have no interest in this promising research line. However, many data suggest that peptides play a crucial role in cancer development, and that peptide receptor antagonists or peptides coupled to cytotoxic agents could be used as antitumor agents opening up new clinical applications in oncology. In this sense, the use of peptides as radiopharmaceuticals for the diagnosis/treatment of NKA-, NKB-, AM-, AM2-, AMY-, and CGRP-positive tumors must be developed. Henceforth, targeted radionuclide cancer therapy is a promising line of research; in fact, peptide receptor-targeted radioligand molecules are being used in research or clinical trials [371,372,373].

The knowledge of the involvement of the SP/NK-1R system in cancer development has notably increased in the last years, and, thanks to this knowledge, the repurposing of the NK-1R antagonist aprepitant has been proposed as an antitumor agent [1,5]. By contrast, there is much to investigate regarding the involvement of NKA, NKB, AM, AM2, AMY, and CGRP in cancer progression. Regarding these peptides, basic information is lacking, and it is scarce or incomplete in many tumors. Peptide receptor antagonists could be used as an antitumor strategy; however, although NK-1R/NK-2R/NK-3R antagonists were safe in clinical trials, they were abandoned due to a lack of efficacy. The ineffectiveness of these antagonists could be due to the low dose used and to a lack of knowledge regarding the molecular interaction between NK receptors and their tachykinin ligands. More systematic research must be developed, and much more basic information is required to attain a similar degree of knowledge regarding the NKA/NK-2R and NKB/NK-2R systems currently on SP/NK1. For example, in vivo studies must be carried out, in vitro investigations combining chemotherapeutic agents and NK-2R/NK-3R antagonists must be performed, the use of NK-2R/NK-3R antagonists as antitumor agents must be widely developed, the roles played by NKA/NKB in the tumor microenvironment must be elucidated, the signaling pathways involved in tumor cell proliferation must be studied in depth, it must be demonstrated whether NK-2R and NK-3R are involved or not in the viability of tumor cells, the use of NKA/NKB and their receptors as cancer predictive factors must be established, and the structure–function relationships between NKA-NK-2R and NKB-NK-3R for the design of new broad-spectrum antitumor drugs must be determined. In this sense, the inhibition of signaling pathways common to several peptides can be a proper antitumor strategy, and it will help to develop broad-spectrum antagonists. Knowing which cancer receptors are involved is crucial to creating specific antitumor ligands and drug-design studies. Thus, understanding how SP, NKA, and NKB and their receptors interact and regulate tumor cells will help to establish specific antitumor strategies.

The involvement of the CT/CGRP peptide family in cancer is more studied than that of NKA/NK-2R and NKB/NK-3R systems; however, it can be said that the research lines that must be developed in the future are, in general, similar to those indicated above for the neurokinin system. More systematic research must be designed to avoid the incomplete knowledge that currently is known regarding the involvement of some peptides belonging to the CT/CGRP peptide family in cancer. There is so much to do. In particular, research lines using peptide receptor antagonists (e.g., AM2 receptor antagonists, NSC-16311 and NSC-37133), peptide receptor interference, peptide fragments (e.g., AM_22–52_), small molecules (e.g., 145425), or neutralizing antibodies against peptides or their receptors (e.g., adrecizumab, mAb-C1, and MoAb-G6) must be potentiated. Treatments against metastatic, angiogenic, and lymphangiogenic mechanisms must also be developed, as well as those increasing the chemosensitivity of tumor cells. Studies focused on the pathological significance of single-nucleotide polymorphisms must also be performed in patients with cancer, since carriers showing the single-nucleotide polymorphism rs4910118 close to the *AM* gene were protected against cancer.

In sum, the involvement of the peptidergic systems in cancer reviewed here must be reopened, developed, and potentiated in the future to attract more researchers to this exciting field and increase interest in the pharmaceutical industry. The data reported in this review indicate that NKA, NKB, AM, AM2, AMY, and CGRP and their receptors are promising antitumor targets, and that the study of the participation of these peptides in cancer development must be significantly increased in future studies.

## Figures and Tables

**Figure 1 cancers-15-01694-f001:**
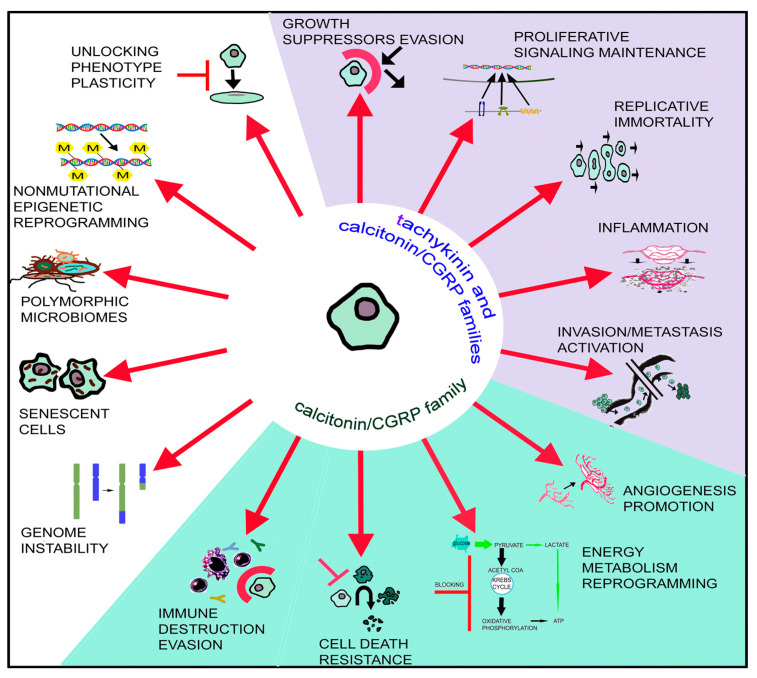
The 14 hallmarks currently considered to be responsible for the development of cancer are indicated. Five hallmarks (blue background) are mediated by peptides belonging to the tachykinin and calcitonin/CGRP peptide families, and four hallmarks (green background) are mediated by peptides belonging to the calcitonin/CGRP peptide family.

**Figure 3 cancers-15-01694-f003:**
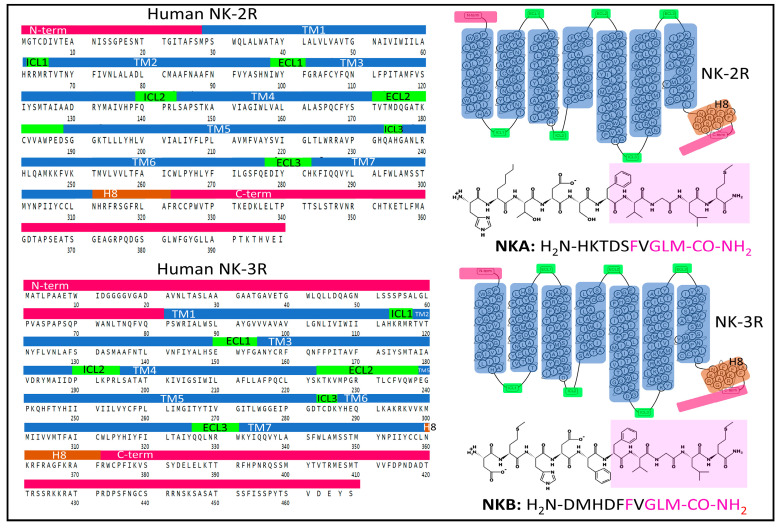
Complete sequences and snake plots of the neurokinin receptors 2 and 3 and their preferential endogenous agonists, NKA and NKB. The domains of NK-2R and NK-3R (N- and C-terminal regions, transmembrane helices (TM), Helix 8, and intra- and extracellular loops) are highlighted in color. Color areas illuminate the amino-acid sequence of the C-terminal end of neurokinins. The sequences and structures of the receptors are from the GPCR database [47], and the peptide structures were drawn with KingDraw software (Version 1.1.0) [48].

**Figure 6 cancers-15-01694-f006:**
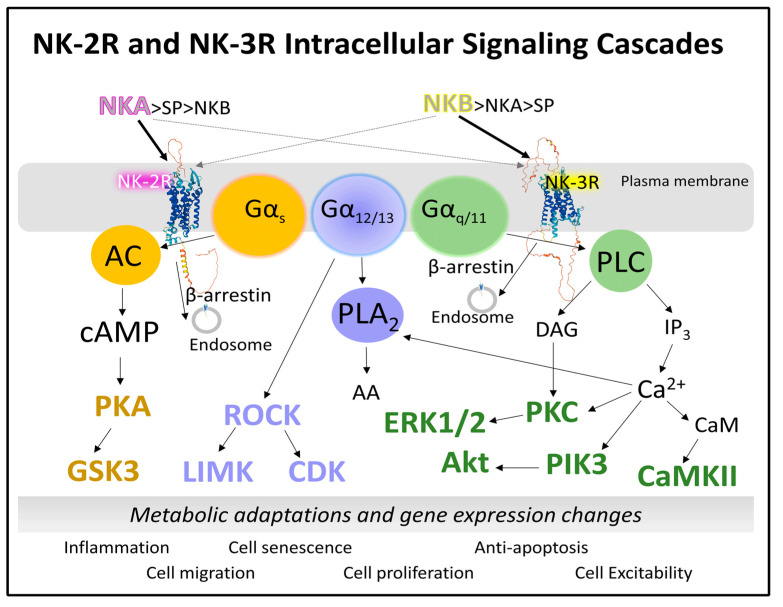
Schematic representation of signaling cascades triggered by NK-2R and NK-3R. Both receptors may recruit Gα subunits to activate phospholipase C, adenylyl cyclase, or phospholipase A2. Consequently, phosphorylation reactions lead to rapid metabolic changes, the appearance of inflammatory messengers, cell proliferation, cell migration, and the arrangement of cytoskeletal proteins or cell excitability (see text for details). Abbreviations: AA, arachidonic acid; AC, adenylyl cyclase; Akt, Ak strain transforming protein kinase; cAMP, cyclic 3′-5′ adenosine monophosphate; CDK, cyclin-dependent kinase; ERK, extracellular signal-regulated receptor kinase; DAG, diacylglycerol; GSK3 (glycogen synthase kinase); IP3, inositol 1,4,5-trisphosphate; LIMK, LIM protein domains (LIN-11, Isl-1, and MEC-3) kinase; NKA, neurokinin A; NKB, neurokinin B; PLA2, phospholipase A2; PKA, protein kinase A; PKC, protein kinase C; PLC, phospholipase C; ROCK, Rho (Ras-homologous)-associated coiled-coil kinase.

**Figure 7 cancers-15-01694-f007:**
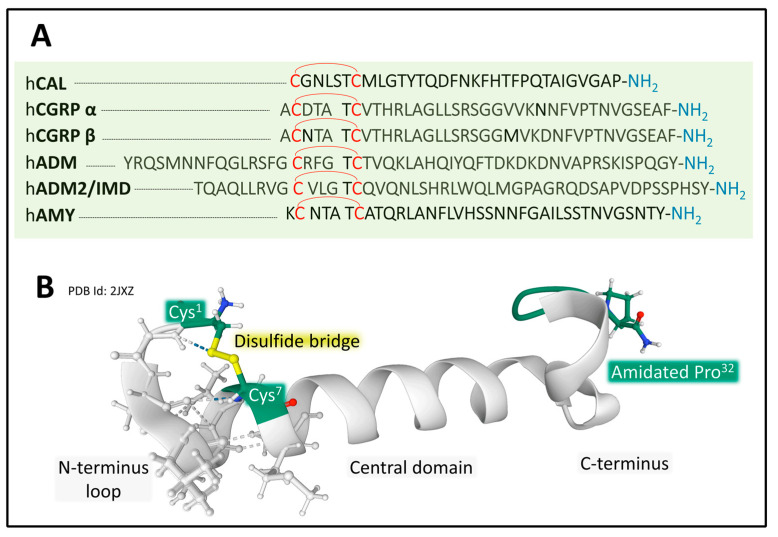
(**A**) shows the amino-acid sequences of the calcitonin/CGRP family. All six peptides present a disulfide bridge between two cysteines in the N-terminal region. Amino-acid sequences are from the GPCR database GPCR database [47]. (**B**) depicts the NMR structure of a human analog of human calcitonin in sodium dodecyl sulfate micelle [115]. Shared structural features include a disulfide bride at the N-terminal region, a helix structure in the central domain, and an amidated C-terminal residue. The configuration corresponding to PDB ID 2JXZ is from the Protein Data Bank [65], drawn with free web-based Mol* software [71].

**Figure 8 cancers-15-01694-f008:**
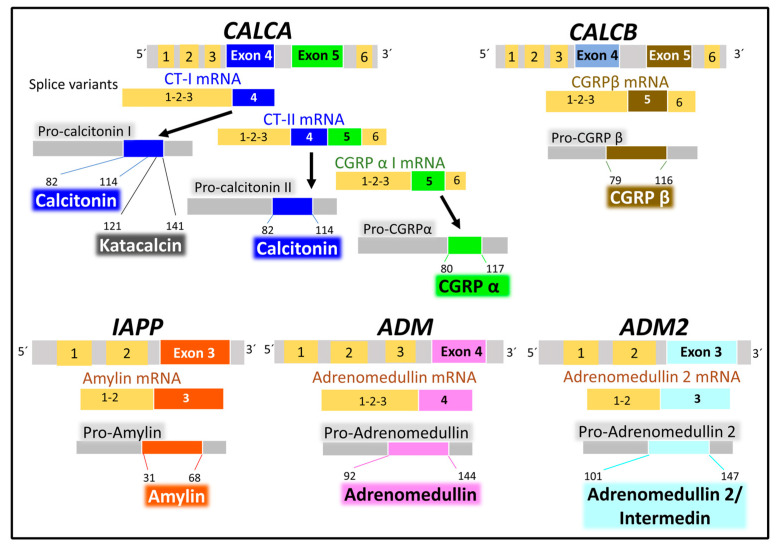
The figure represents the human calcitonin/CGRP family genes and their main peptide products, calcitonin, CGRP (calcitonin gene-related peptide) isoforms α and β, amylin, adrenomedullin, and adrenomedullin 2/intermedin. Alternative splicing generates mRNA isoforms transcribed into peptides that undergo post-translational modifications (protein hydrolysis, disulfide bridge formation, and amination of C-terminal amino acid), giving rise to functional peptides. Data supporting this scheme are from the UNIPROT [25] and Entrez Gene (National Library of Medicine) [34] databases. Numbers indicate the residue position within the peptide sequences.

**Figure 9 cancers-15-01694-f009:**
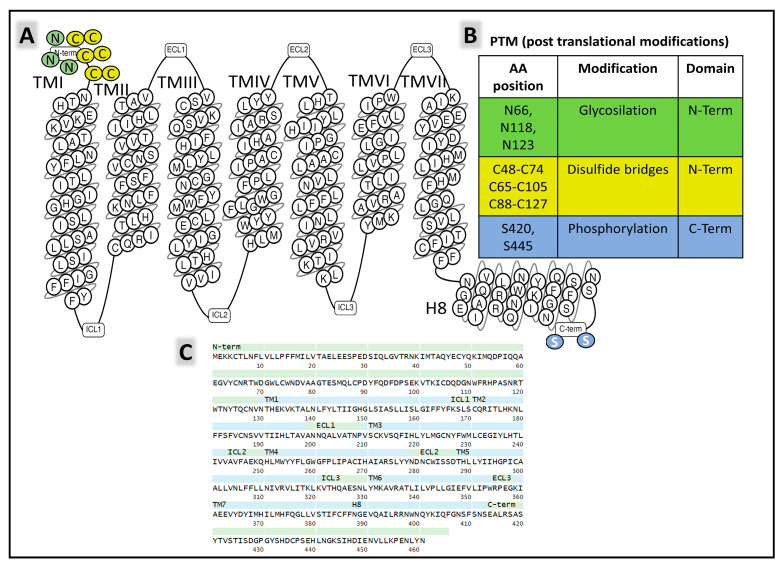
(**A**) is a snake plot [47] of the general architecture of CLR (calcitonin receptor-like receptor), indicating transmembrane segments and intra- (ICL) and extracellular (ECL) loops. (**B**) displays amino acids undergoing post-translational modifications [25]. (**C**) shows the primary structure of the receptor and the sequences forming different protein domains [47], TM (transmembrane), ECL (extracellular loops), ICL (intracellular loops), and H8 (helix 8).

**Figure 10 cancers-15-01694-f010:**
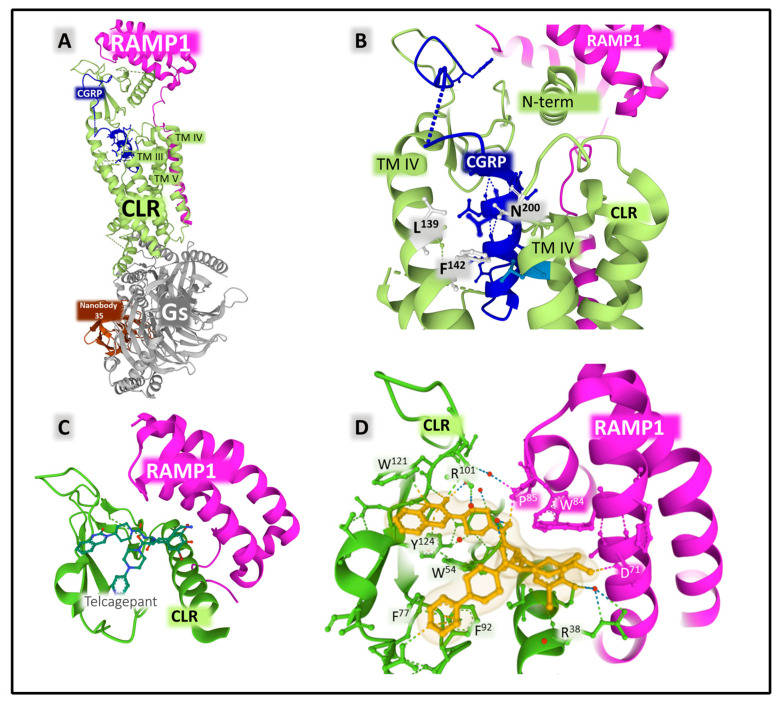
(**A**) shows the crystal complex formed by CLR (calcitonin receptor-like receptor in green) bound to protein RAMP1 (receptor activity modifying protein, in pink), Gs heterotrimeric protein (gray), and nanobody 35 (brown). The interaction of RAMP 1 with the transmembrane segments III, IV, and V and the N-terminal portion of CLR are indicated in the figure. (**B**) depicts the binding site of CGRP (calcitonin gene-related peptide, blue), showing the amino-acid positions of CLR contacting the agonist peptide. (**C**) shows the crystal structure of an ectodomain complex of the CGRP receptor bound to the antagonist telcagepant (MK0974). (**D**) highlights the contacts between the antagonist and the receptor structure. The figures mark the position of transmembrane (TM) helices. Dashed lines indicate weak interactions stabilizing peptide structure. The representations in (**A**,**B**) corresponding to PDB ID 6E3Y [123], and those in (**C**,**D**) corresponding to PDB ID 3N7S [127] were obtained from the Protein Data Bank (PDB [65]) and colored with Mol* free web-based software [71].

**Figure 11 cancers-15-01694-f011:**
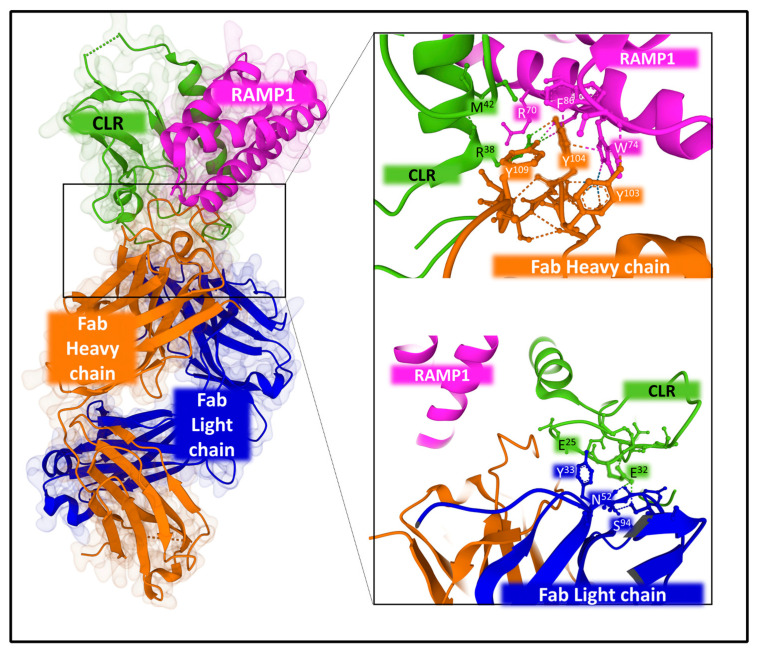
Structure of the complex CLR-RAMP1 co-crystalized with monoclonal antibody 6UMG. The figure on the right-hand side depicts some atomic contacts of the antibody heavy chain (upper panel) and the antibody light chain (lower panel) with the interface of the complex CLR-RAMP. The structures corresponding to PDB ID 6UMG [129] were obtained from the Protein Data Bank (PDB [65]) and designed with Mol* free web-based software [71].

**Figure 12 cancers-15-01694-f012:**
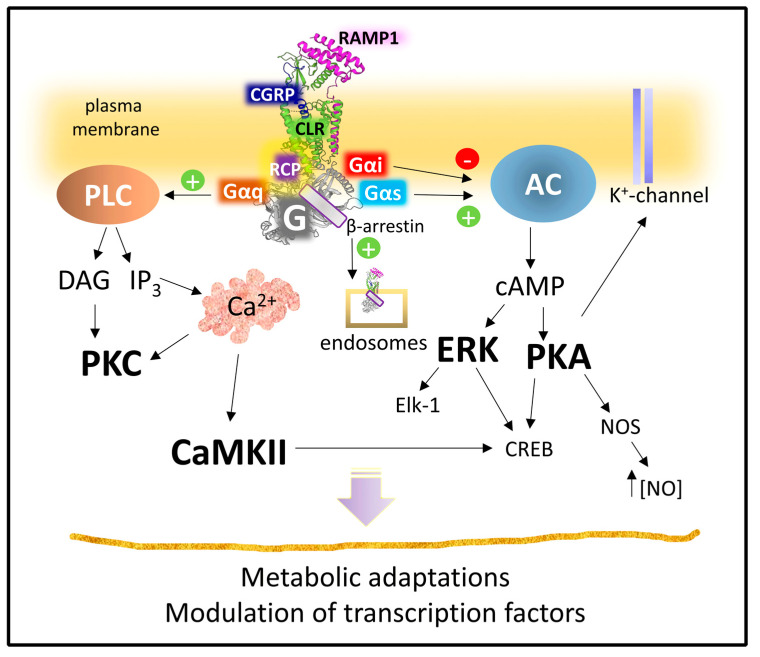
Signaling pathways activated by CGRP receptor. Abbreviations: CaMKII, calcium-calmodulin kinase II; cAMP, cyclic adenosine 5′monophosphate; CLR, calcitonin receptor-like receptor; CREB, cAMP-response element binding protein; DAG, diacylglycerol; Elk-1, ETS-erythroblast transformation specific-like protein; ERK, extracellular receptor kinase; IP3, inositol 1,4,5-trisphosphate; NOS, nitric oxide synthase; PKA, protein kinase A; PKC, protein kinase C; RAMP1, receptor activity modifying protein 1; RCP, receptor component protein (see text for details).

**Figure 13 cancers-15-01694-f013:**
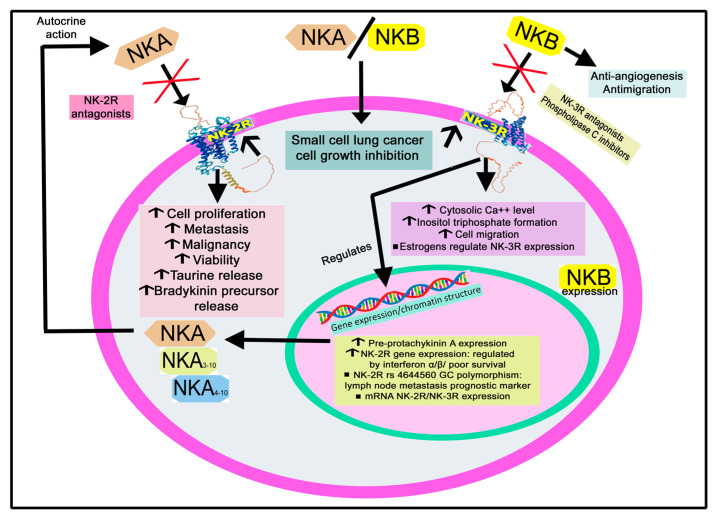
Summary of the mechanisms mediated by NKA/NKB via NK-2R/NK-3R in tumor cells. NKA: neurokinin A; NKB: neurokinin B; NK-2R: neurokinin-2 receptor; NK-3R: neurokinin-3 receptor. ↑: increase; ↓: decrease.

**Figure 14 cancers-15-01694-f014:**
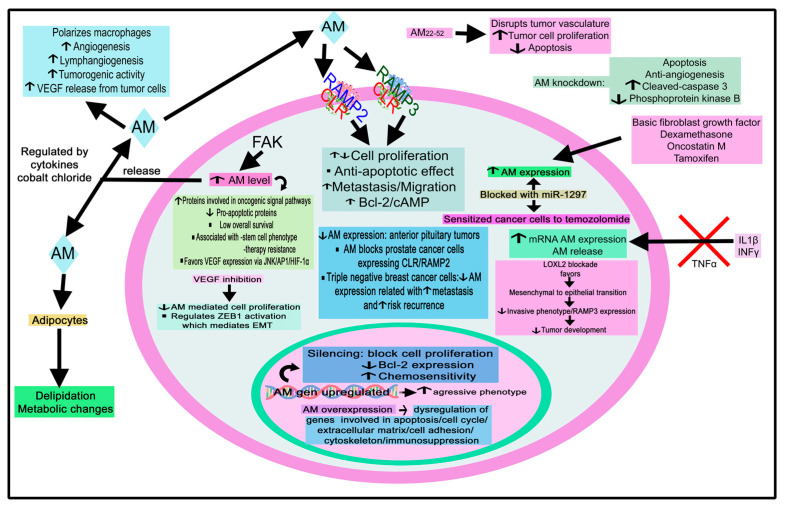
Summary of the mechanisms mediated by adrenomedullin (AM) in tumor cells. AP1: activator protein 1; cAMP: cyclic 3′-5′ adenosine monophosphate; CLR: calcitonin receptor-like receptor; EMT: epithelial-to-mesenchymal transition; FAK: focal adhesion kinase; JNK: c-Jun N-terminal kinase; INF γ: interferon γ; IL-1β: interleukin 1β; RAMP2/3: receptor activity modifying protein 2 and 3; VEGF: vascular endothelial growth factor. ↑: increase; ↓: decrease.

**Figure 15 cancers-15-01694-f015:**
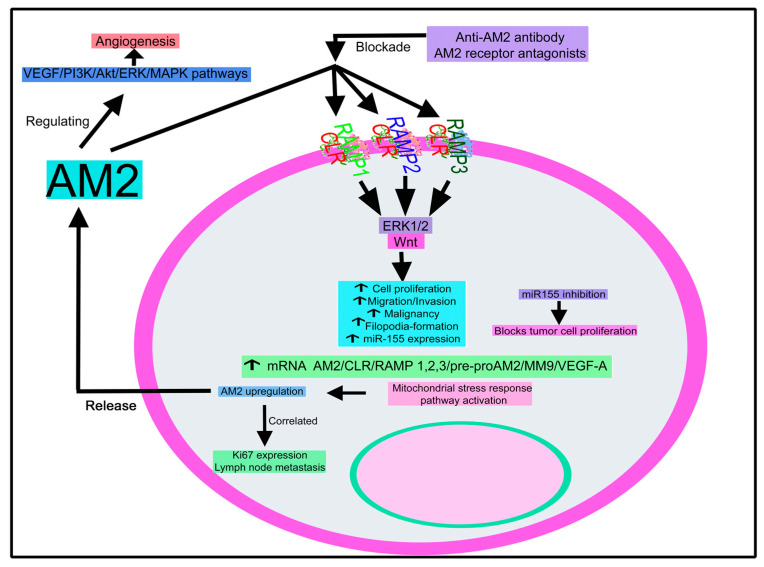
Summary of the mechanisms mediated by adrenomedullin 2 (AM2) in tumor cells. Akt: Ak strain transforming protein kinase; CLR: calcitonin receptor-like receptor; ERK 1/2: extracellular signal-regulated protein kinase; MAPK: mitogen-activated protein kinase; MMP9: matrix metalloproteinase 9; PI3K: phosphatidylinositol 3-kinase; RAMP1/2/3: receptor activity modifying protein 1, 2 and 3; VEGF-A: vascular endothelial growth factor A. ↑: increase; ↓: decrease.

**Figure 16 cancers-15-01694-f016:**
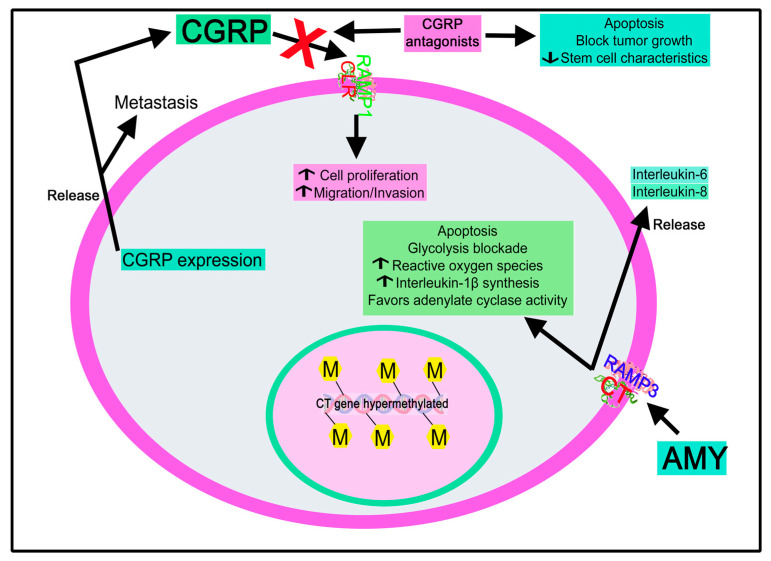
Summary of the mechanisms mediated by amylin (AMY) and calcitonin gene-related peptide (CGRP) in tumor cells. CLR: calcitonin receptor-like receptor; RAMP1/3: receptor activity modifying proteins 1 and 3. ↑: increase; ↓: decrease.

**Table 1 cancers-15-01694-t001:** Involvement of NKA/NKB in cancer.

Tumor	NKA/NKB	References
Breast cancer	Malignant biopsies/cancer cells: increased pre-protachykinin A expression. Metastatic cancer cells: NK-2R overexpression; NKA increases aggressiveness and NK-2R expression. NKA favors cell proliferation. NK-2R mediates cancer cell proliferation but not in normal cells. NK-2R antagonists block tumor cell proliferation.	[92,139,140,141]
Colorectal cancer	High *NK-2R* gene expression relates to poor survival. NK-2R rs4644560 GC polymorphism alone or combined with NK-1R rs10198644 GC: a prognostic marker for lymph node metastasis. NK-2R overexpression: increases tumorigenesis and metastatic colonization. NKA increases viability and tumor cell proliferation. NK-2R antagonists decrease tumor cell proliferation.	[142,143]
Glioma	NKA promotes tumor cell proliferation through NK-1R. Astrocytoma: NK-3R expression.	[144,145]
Insulinoma	Cancer cells release NKA and express the *pre-protachykinin A* gene.	[146,147]
Lung cancer	mRNA pre-protachykinin/NKA expression. NKA/NKB blocks tumor cell growth.	[20,148]
Midgut carcinoid tumor	High plasma NKA level: worse survival. Plasma NKA level: a biomarker of prognosis. Pre-pro-tachykinin I mRNA/NKA expression. NKA/NKB increased luminal content.	[148,149,150,151]
Neuroblastoma	mRNA NK-2R/NK-3R expression. NKB but not NKA expression. NK-2R/NK-3R: activated independently by NKA. NK-2R mediates cell proliferation. NKA, via NK-3R, increases Ca^++^ cytosolic level through phospholipase C activation; this action is blocked with SR-48968.	[152,153,154,155]
Oral squamous cell carcinoma	High NK-3R expression. NKB expression was not observed. B-222,200 inhibits tumorigenesis	[21,156]
Phaeochromocytoma	Human phaeochromocytoma extract: NKA expression. NKB was observed in one of 10 phaeochromocytomas.	[31,157]
Schwannoma-derived cells	NKA expression.	[158]
Small bowel neuroendocrine tumors	Plasma NKA level: biomarker. High levels are associated with poor prognosis. Improved prognosis by lowering plasma NKA level. Ileal metastatic carcinoid: circulating NKA increase. NKA, NKA_3–10_, and NKA_4–10_ expression.	[159,160,161,162,163,164,165]
Uterine leiomyomata	NK-2R mRNA upregulation. High NKB/NK-3R expression. NK-3R activation: nuclear translocation affecting gene expression/chromatin structure.	[91,166]

**Table 2 cancers-15-01694-t002:** Participation of adrenomedullin (AM), adrenomedullin 2 (AM2), amylin (AMY), and calcitonin gene-related peptide (CGRP) in cancer.

Tumor	AM/AM2/AMY/CGRP	References
Acute myeloid leukemia	AM, CLR, RAMP 2/3 expression. A high AM level is associated with low overall survival/disease-free survival. AM/AM_22–52_ regulates cell growth. AM expression associated with genes related to immunosuppression resistance. AM correlates with adverse outcomes. Targeting calcitonin receptor-like receptors prevents relapse. *Calcitonin* gene methylation pattern: independent prognostic factor. High calcitonin receptor expression: poor prognosis and correlates with chemotherapy resistance. Olcegepant decreases leukemic burden/stem cell characteristics. MK0974 promotes apoptosis.	[136,182,183,184,185,186]
Adrenocortical tumor	AM is synthesized/released from adrenocortical tumors and phaeochromocytomas. Phaeochromocytomas: higher AM/receptor mRNA expression. Phaeochromocytomas and adenomas: AM2 expression. AM blocks phaeochromocytoma cell proliferation. AM level as a biomarker. Adrenal tumors: AM2, CLR, RAMP1/RAMP2/RAMP3 mRNA expression. Phaeochromocytomas: high CGRP tissue level.	[187,188,189,190,191,192]
Bladder cancer	High AM level. AM knockdown promotes apoptosis. AM knockdown/cisplatin combination decreases tumor growth.	[193]
Breast cancer	AM expression associated with axillary lymph node metastasis. RAMP3: involved in metastasis. Tumor cells overexpressing AM: potential angiogenic increase/less apoptotic mechanisms. Tumor cells expressing/releasing AM: favor cell proliferation, breast cancer bone metastasis, and angiogenesis. AM is a tumor survival factor. Triple-negative breast cancer samples: AM expression decreased; this low expression is related to poor prognosis and increased risk of recurrence/metastasis. AM_22–52_ disrupts tumor vasculature, decreases tumor cell proliferation, and induces apoptosis. Plasma AM2 level associated with poor patient outcomes. AM2 expression increased: correlated with Ki67 expression/lymph node metastasis. AM2 promotes cancer cells’ growth, migration, and invasion; these actions were blocked with anti-AM2 antibodies. CGRP expression increased. CGRP is involved in metastasis.	[99,194,195,196,197,198,199,200,201]
Choriocarcinoma	AM mRNA expression.	[202]
Colon cancer	CLR and RAMP2/3 expressions. High levels of AM, CLR, and RAMP2/3 are correlated with lymph nodes and distant metastasis. High AM level is related to low disease-free survival. Higher AM level and AM mRNA expression. AM promotes tumor cell proliferation/invasion and antiapoptotic effect. AM level associated with clinical survival rate/cancer stage. Knockdown AM promotes apoptosis/blocks angiogenesis. AM expression is associated with vascular endothelial growth factor and hypoxia-inducible factor-1α. AM positive modulator (145425) decreases the number of tumors. Higher mRNA pre-proAM, pre-proAM2, CLR, RAMP2/3, MMP-9, and VEGF-A expression. Positive correlation between *MMP-9* gene expression and pre-proAM, but not pre-proAM2.	[179,203,204,205,206,207,208]
Cutaneous nerve neuromas	Saphenous nerve neuromas: CGRP release from nerve fibers.	[209]
Endometrial cancer	AM favors angiogenesis/tumor growth and blocks tumor cell death. AM level increases from normal, simple, or complex hyperplasia with or without atypia to grade 1 adenocarcinoma.	[101,210,211,212]
Ewing sarcoma	CGRP expression. CGRP promotes the proliferation of cancer cells.	[213]
Gastric cancer	Tumor-derived AM promotes mast cell degranulation, as well as favors tumor cell proliferation, and the apoptosis blockade in cancer cells.	[214]
Glioma	AM favors mitogenesis in tumor cells and exerts angiogenic/antiapoptotic effects. AM mRNA is associated with tumor type and grade. c-Jun/JNK pathway is involved in the growth regulatory activity mediated by AM. AM expression is upregulated in temozolomide-resistant glioma samples: miR-1297 targets AM, blocks its expression, and sensitizes tumor cells to temozolomide treatment. AM2 expression is increased and correlated with higher-grade gliomas. AM2 increases the invasive capacity of tumor cells and improves tumor blood. AMY promotes the release of inflammatory cytokines from tumor cells. The signal transducer and STAT-3 in astroglioma control AM expression. AM promotes the migration of astroglioma cells.	[215,216,217,218,219,220,221,222,223]
Head and neck squamous cell carcinoma	Peripheral nerve terminals release CGRP, which exerts a paracrine action on tumor cells. CGRP links perineural invasion and lymph node metastasis. Pre-operative plasma CGRP level: lymph node metastasis predictor.	[224,225]
Liver cancer	CLR and RAMP2/3 expression. High AM level is related to increased intrahepatic metastasis. Higher AM mRNA levels in tumor tissues than in adjacent nontumor tissues. AM mediates the epithelial–mesenchymal transition and promotes tumor cell growth. Knockdown AM expression promotes apoptotic mechanisms and, combined with cisplatin, decreases tumor growth. Microvessel density and mRNA AM/erythropoietin receptor levels are higher in hepatocellular carcinoma than in nontumor tissues: both levels are correlated with tumor metastasis, pathological differentiation, and capsule invasion. AM is associated with N-cadherin intensity, vascular invasion, and poor prognosis. AM level as a prognostic factor. High AM2 mRNA expression even in early stages. AM2 increases tumor cell proliferation and survival, and is involved in angiogenesis: AM2_17–47_ blocked this proliferation. AMY binding site expression.	[226,227,228,229,230,231,232,233,234]
Lung cancer	AM expression does not correlate with survival, cancer stage, or tumor differentiation. AM contributes to the carcinogenicity of tobacco-activated aryl hydrocarbon receptor products. *CGRP* gene expression.	[235,236,237,238]
Melanoma	Tumor-associated macrophages, through AM, favor melanoma growth and angiogenesis. The number of tumor-associated macrophages (express/release AM) in the tumor microenvironment is correlated with poor prognosis. Tumor-associated macrophages favor the migration of endothelial cells and increase tumor cell growth.	[239]
Nasopharyngeal carcinoma	AM level as a biomarker for predicting prognosis.	[240]
Neuroblastoma	AM receptor expression. AM mRNA expression is associated with tumor differentiation.	[241,242]
Neuroendocrine tumors	High plasma and tissue AM expression: predictive factors for tumor progression and worsened prognosis.	[243]
Oropharyngeal squamous cell carcinoma	Jumonji domain-containing 1A, H3K9me1/2, and AM expression: predictor markers for progression and prognosis. *JMJD1A* gene target AM favors tumorigenesis/cell growth.	[244]
Osteosarcoma	AM expression is associated with metastasis degree/malignancy. AM overexpression. AM exerts an antiapoptotic effect.	[245,246]
Ovarian cancer	High AM level is related to tumor stage. *AM* gene is correlated with histological grade, lymph node metastasis, and prognosis but not with disease stage, histological subtype, residual tumor mass after initial surgery, and patient’s age at diagnosis. Tumor cells express AM mRNA for both ligand/receptor. AM is involved in tumor progression/cell migration. *AM* gene silencing blocks cell proliferation and increases tumor cell chemosensitivity. Cancer patients with high AM expression show larger residual size of tumors, shorter disease-free/overall survival time, and higher metastasis incidence. AM as biomarker to evaluate prognosis/malignant potential. AM is correlated with expressions of HIF-1α, VEGF, or microvessel density. AM favors angiogenesis.	[247,248,249,250,251,252]
Pancreatic cancer	High AM levels are associated with disease-free survival decrease. Higher AM plasma level. AM level is higher in cancer patients with diabetes than those without diabetes. Insulinoma: circulating AM increased. AM and its receptor are expressed. AM is involved in tumor cell proliferation, migration, invasion, metastasis, and angiogenesis. AM mRNA/protein expressions are increased. AM as a tumor marker. AM receptor silencing blocks AM-induced cell growth and invasion. AM antagonists decrease tumor cell growth/blood vessel diameter. Selective RAMP2 activation/RAMP3 inhibition: tumor metastasis suppression. AM2 as tumor angiogenic factor. AM2 level: poorer survival predictor. AM2 is a biomarker predicting survival. Insulinomas express AMY. Plasma AMY levels are increased in nondiabetic patients with cancer; this level is low in patients with diabetes. CALCA (αCGRP) and CALCB (βCGRP) methylation increases.	[253,254,255,256,257,258,259,260,261,262,263,264]
Pituitary adenoma	AM expression is decreased in anterior pituitary tumors.	[265]
Prostate cancer	AM is expressed in prostate carcinomas. A high AM level is associated with a high Gleason score. Plasma AM2 level is associated with Gleason’s score, tumor node metastasis, and 5 year metastasis. AM overexpression inhibits tumor cell growth and dysregulates genes involved in apoptosis, cell cycle, extracellular matrix, cell adhesion, and cytoskeleton. AM promotes human prostate growth via the AM2 receptor (CLR/RAMP3) subtype. AM blocks apoptosis in specific tumor cells. AM, upon androgen ablation, is involved in hormone-independent tumor growth, lymphangiogenesis, and neoangiogenesis. AM promotes cancer cell migration and invasion. AM2 is involved in cancer cell migration/angiogenesis. Higher plasma AM2 level in patients with prostate cancer. Patients with a Gleason score ≥7, unconfined organ, seminal vesicle invasion, tumor node metastasis stage T2, positive lymph node, or extra-prostatic extension show high AM2. AM2 as a prognostic, predictive biomarker for 5 year metastasis and 5 year progression. CGRP increases the invasive/migratory capacity of tumor cells. CGRP serum level correlates with cancer progression. Higher serum CGRP level is associated with higher histological grade/clinical stages. CGRP receptor mediates metastasis. CGRP promotes tumor growth which is blocked with CGRP antagonists.	[180,266,267,268,269,270,271,272,273,274,275]
Renal carcinoma	AM, CLR, and RAMP2/3 expression. High CLR level is associated with high tumor grade. AM and AM mRNA expression. AM mRNA expression is correlated with VEGF-A mRNA. AM, via CLR/RAMP2 and CLR/RAMP3 receptors, promotes cell proliferation, migration, and invasion. High AM mRNA level is related to increased risk of relapse.	[276,277,278]
Thymic lymphomas	Pramlintide promotes tumor regression. AMY blocks glycolysis, promotes reactive oxygen species formation, and induces apoptosis.	[279]
Thyroid cancer	AM2 expression is increased in obese patients with thyroid cancer, showing locoregional recurrence, a high prevalence of lymph node metastasis, and larger tumor size. High circulating AM2/tumor cell AM2 expression levels are associated with aggressive pathological parameters. AM2 as a biomarker for predicting thyroid tumor progression. Medullary thyroid carcinoma: high AMY levels. Plasma AMY/insulin levels are correlated in medullary thyroid carcinoma. *CGRP* gene expression. Plasma CGRP level marker for medullary thyroid carcinoma.	[237,280,281,282,283]
Uterine cervical carcinoma	High AM and AM mRNA expression. AM is involved in promoting malignant progression and in selecting carcinoma cells resistant to apoptosis.	[284,285]

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
