# Peer review of "Peptidergic Systems and Cancer: Focus on Tachykinin and Calcitonin/Calcitonin Gene-Related Peptide Families"

_cancers, 2023, doi:10.3390/cancers15061694_

Round 1
Reviewer 1 Report
Dear Editor,
This manuscript, that is about the role of tachykinins and calcitonin/calcitonin gene-related peptides in carcinogenesis, is a comprehensive review which is very well designed and well written. The text is very easy to follow and figures are attractive and informative.
With respect to the expertise of the authors in this field, I recommend the publication of this paper in the present format.
Best regards
Author Response
This manuscript, that is about the role of tachykinins and calcitonin/calcitonin gene-related peptides in carcinogenesis, is a comprehensive review which is very well designed and well written. The text is very easy to follow and figures are attractive and informative.
With respect to the expertise of the authors in this field, I recommend the publication of this paper in the present format.
Thank you very much for your positive comments.

Reviewer 2 Report
The review paper entitled "Peptidergic Systems and Cancer: Focus on Tachykinin and Calcitonin/Calcitonin Gene‐Related Peptide Families" provides a comprehensive overview of the role of peptidergic systems in cancer and anti-cancer therapy. The authors have presented a thorough overview of the structure, functionality, and cell signaling pathways associated with tachykinin and calcitonin/calcitonin gene-related peptide families in different types of cancer.
The paper is well-written and organized, with clear subheadings and self-explanatory figures that make it easy to follow. The tables provided are also very useful, as they summarize the participation of peptides in different cancer types. One suggestion for improvement would be to trim some of the content that is less focused on cancer. While the authors provide a comprehensive overview of peptidergic systems in general, some of the details may not be directly relevant to cancer. By focusing more on the specific ways in which peptidergic systems are involved in cancer and anti-cancer therapy, the paper could be even more informative and useful to readers.
Overall, the paper provides a comprehensive and valuable resource for understanding the involvement of peptidergic systems in cancer. I recommend this paper for publication with minor revisions as suggested, and I believe it will be of great interest to researchers and clinicians in the field.
Minor Comments:
1. Add a definition or brief introduction of peptidergic systems in the “Introduction” section.
2. Add citations at the end of the sentence “Peptides such as substance P (a full review focused on its participation in cancer has recently been published [5]), neurotensin, orexin, angiotensin II, neuropeptide Y, vasoactive intestinal peptide calcitonin gene‐related peptide, adrenomedullin, adrenomedullin 2 or intermedin, and amylin contribute to cancer development” at line 43 to 47.
3. At line 47-48, the author mentioned “These peptides promote the mitogenesis/migration of tumor cells, exert an antiapoptotic action, and stimulate the growth of blood vessels and lymphangiogenesis.” However, several peptides, such as neuropeptide Y, orexin and vasoactive intestinal peptide, have been shown to have both pro- and anti-cancer effects. (Wu, Y., Berisha, A., Borniger, J.C., Neuropeptides in Cancer: Friend and Foe? Adv. Biology 2022).
4. In the legend of Figure 1 at line 87-88, “Five hallmarks (blue background) are mediated by peptides belonging to the tachykinin or calcitonin/CGRP peptide families”, it is better to change “or” into “and” to be consistent with the description in the figure.
5. The paragraph spanning from line 92 to 106, and line 1450-1462 can be trimmed more relevant to cancer and eliminate repetitive information.
6. In Figure 3, the green highlight label for ICL3 in NK-2R is hidden (top-right panel).
7. The content spanning from line 413 to 431 is not cancer-related, can be trimmed down or removed.
8. In the “Modification” column of Figure 9B, replace “Disulphide bridge” into “Disulfide bridges” to maintain the same spelling throughout the context.
9. At line 739, change “RAMP2, and 3” to “RAMP2 and RAMP3” to make it clear.
10. In Figure 15, the mechanism of AM2 in tumor cell nucleus seems missing.
11. As mentioned in line 1790-1793, “In this sense, the use of peptides as radiopharmaceuticals for the diagnosis/treatment of NKA‐, NKB‐, AM‐, AM2‐, AMY‐, and CGRP‐positive tumors must be developed. Henceforth, targeted radionuclide cancer therapy is a promising line of research.” Yes, some of these peptide receptor-targeted radioligand molecules have been developed and used in research or clinical trials (e.g. NK1R radioligands, Majkowska-Pilip, A., Halik, P. K., & Gniazdowska, E. (2019); Radiolabeled CGRP antagonists, patent WO2011014383A1; radionuclide targeting VIP receptor, Tang, B., Yong, X., Xie, R., Li, Q. W., & Yang, S. M. (2014)).
Author Response
The review paper entitled "Peptidergic Systems and Cancer: Focus on Tachykinin and Calcitonin/Calcitonin Gene‐Related Peptide Families" provides a comprehensive overview of the role of peptidergic systems in cancer and anti-cancer therapy. The authors have presented a thorough overview of the structure, functionality, and cell signaling pathways associated with tachykinin and calcitonin/calcitonin gene-related peptide families in different types of cancer.
The paper is well-written and organized, with clear subheadings and self-explanatory figures that make it easy to follow. The tables provided are also very useful, as they summarize the participation of peptides in different cancer types. One suggestion for improvement would be to trim some of the content that is less focused on cancer. While the authors provide a comprehensive overview of peptidergic systems in general, some of the details may not be directly relevant to cancer. By focusing more on the specific ways in which peptidergic systems are involved in cancer and anti-cancer therapy, the paper could be even more informative and useful to readers.
Overall, the paper provides a comprehensive and valuable resource for understanding the involvement of peptidergic systems in cancer. I recommend this paper for publication with minor revisions as suggested, and I believe it will be of great interest to researchers and clinicians in the field.
Thank you very much for your positive comments. According to the suggestions of the reviewer the new version has been adjusted as follows (changes highlighted in yellow):
Minor Comments:
- Add a definition or brief introduction of peptidergic systems in the “Introduction” section. This has been done. See page 1.
- Add citations at the end of the sentence “Peptides such as substance P (a full review focused on its participation in cancer has recently been published [5]), neurotensin, orexin, angiotensin II, neuropeptide Y, vasoactive intestinal peptide calcitonin gene‐related peptide, adrenomedullin, adrenomedullin 2 or intermedin, and amylin contribute to cancer development” at line 43 to 47. This has been done. See page 2.
- At line 47-48, the author mentioned “These peptides promote the mitogenesis/migration of tumor cells, exert an antiapoptotic action, and stimulate the growth of blood vessels and lymphangiogenesis.” However, several peptides, such as neuropeptide Y, orexin and vasoactive intestinal peptide, have been shown to have both pro- and anti-cancer effects. (Wu, Y., Berisha, A., Borniger, J.C., Neuropeptides in Cancer: Friend and Foe? Adv. Biology2022). This reference (number 7) has been added. See pages 2 and 52. Thus, references have been renumbered.
- In the legend of Figure 1 at line 87-88, “Five hallmarks (blue background) are mediated by peptides belonging to the tachykinin or calcitonin/CGRP peptide families”, it is better to change “or” into “and” to be consistent with the description in the figure. This has been corrected. See page 3.
- The paragraph spanning from line 92 to 106, and line 1450-1462 can be trimmed more relevant to cancer and eliminate repetitive information. Both paragraphs have been shortened and the text focused on cancer. See pages 3 and 44.
- In Figure 3, the green highlight label for ICL3 in NK-2R is hidden (top-right panel). This has been corrected. See page 5.
- The content spanning from line 413 to 431 is not cancer-related, can be trimmed down or removed. This paragraph has been shortened and several references deleted. See page 12.
- In the “Modification” column of Figure 9B, replace “Disulphide bridge” into “Disulfide bridges” to maintain the same spelling throughout the context. This has been corrected. See Figure 9, page 14.
- At line 739, change “RAMP2, and 3” to “RAMP2 and RAMP3” to make it clear. Done. See page 30.
- In Figure 15, the mechanism of AM2 in tumor cell nucleus seems missing. No known nuclear activity; thus, nothing to indicate inside the nucleus. See page 43.
- As mentioned in line 1790-1793, “In this sense, the use of peptides as radiopharmaceuticals for the diagnosis/treatment of NKA‐, NKB‐, AM‐, AM2‐, AMY‐, and CGRP‐positive tumors must be developed. Henceforth, targeted radionuclide cancer therapy is a promising line of research.” Yes, some of these peptide receptor-targeted radioligand molecules have been developed and used in research or clinical trials (e.g. NK1R radioligands, Majkowska-Pilip, A., Halik, P. K., & Gniazdowska, E. (2019); Radiolabeled CGRP antagonists, patent WO2011014383A1; radionuclide targeting VIP receptor, Tang, B., Yong, X., Xie, R., Li, Q. W., & Yang, S. M. (2014)). These references have been added [371-373]. See pages 51 and 68.
